DOI: 10.1038/s41467-018-07185-y　　OPEN

# Proteome-wide analysis of USP14 substrates revealed its role in hepatosteatosis via stabilization of FASN

Bin Liu[1,2,3], Shangwen Jiang[3,4], Min Li[1], Xuelian Xiong[1], Mingrui Zhu[3,4], Duanzhuo Li[2], Lei Zhao[3], Lili Qian[3,4], Linhui Zhai[3], Jing Li [5], Han Lu[6], Shengnan Sun[3], Jiandie Lin[7], Yan Lu [1], Xiaoying Li[1] & Minjia Tan [3,4]

Ubiquitin-specific protease 14 (USP14) is one of the major proteasome-associated deubi-quitinating enzymes critical for proteome homeostasis. However, substrates of USP14 remain largely unknown, hindering the understanding of its functional roles. Here we conduct a comprehensive proteome, ubiquitinome and interactome analysis for USP14 substrate screening. Bioinformatics analysis reveals broad new potential roles of USP14, especially in lipid and carbohydrate metabolism. Among the potential substrates identified, we show that fatty acid synthase (FASN), a key enzyme involved in hepatic lipogenesis, is a bona fide substrate of USP14. USP14 directly interacts with and increases FASN stability. As a result, overexpression of USP14 promotes liver triglyceride accumulation in C57BL/6 mice, whereas genetic ablation or pharmacological inhibition of USP14 ameliorates hepatosteatosis, hyperglycemia and insulin resistance in obese mice. In conclusion, our findings reveal for the first time an indispensable role of USP14 in hepatosteatosis through FASN stabilization.

[1] Department of Endocrinology and Metabolism, Zhongshan Hospital, Fudan Institute of Metabolic Diseases, Key Laboratory of Metabolism and Molecular Medicine, the Ministry of Education, Fudan University, Shanghai 200032, PR China. [2] Hubei Key Laboratory for Kidney Disease Pathogenesis and Intervention, Hubei Polytechnic University School of Medicine, Huangshi, Hubei 435003, PR China. [3] State Key Laboratory of Drug Research, Shanghai Institute of Materia Medica, Chinese Academy of Sciences, Shanghai 201203, PR China. [4] University of Chinese Academy of Sciences, Beijing, PR China. [5] Department of Bioinformatics and Biostatistics, School of Life Sciences and Biotechnology, Shanghai Jiao Tong University, Shanghai 200240, PR China. [6] Department of Anesthesiology, Ruijin Hospital, Shanghai Jiao-Tong University School of Medicine (SJTU-SM), Shanghai 200025, PR China. [7] Life Sciences Institute and Department of Cell and Developmental Biology, University of Michigan Medical Center, Ann Arbor, MI, USA. These authors contributed equally: Bin Liu, Shangwen Jiang, Min Li. Correspondence and requests for materials should be addressed to Y.L. (email: lu.yan2@zs-hospital.sh.cn) or to X.L. (email: li.xiaoying@zs-hospital.sh.cn) or to M.T. (email: mjtan@simm.ac.cn)

The ubiquitin-proteasome system (UPS) controls intracellular protein degradation through a series of steps, including substrate recognition, ubiquitin conjugation, and ubiquitinated substrates degradation by proteasomes[1]. In mammals, E3 ubiquitin ligases and deubiquitylating enzymes (DUBs) play central roles in protein degradation and turnover through protein ubiquitination and deubiquitination[2]. DUBs catalyze the removal of ubiquitins from their target proteins and render ubiquitin homeostasis a highly dynamic process. There are 79 DUBs encoded by the mammalian genome[3]. Among them, the ubiquitin-specific proteases (USPs) are the largest family of DUBs, which participates in diverse cellular processes, including cell cycle progression, cell proliferation and differentiation, transcriptional regulation, and modulation of plasma membrane receptors[4].

USP14 is the only USP family of DUBs that reversibly associate with the proteasomal 19S regulatory particle[5]. USP14 serves as a quality control component to rescue proteins from degradation by dissembling the ubiquitin chain from its substrate distal tip[6]. Many studies have shown that USP14 plays critical roles in cellular signaling, neurological functions, and tumorigenesis. USP14 is a mediator of Dishevelled (Dvl) deubiquitination for Wnt signaling pathway regulation[7], and inhibit nuclear factor (NF)-κB signaling through NLRC5 deubiquitination[8]. In neurological systems, USP14 was reported to regulate long-term memory formation, while alteration of USP14 contributes to loss-of-mobility and early postnatal lethality in ataxia ($ax^J$) mice[9,10]. USP14 was overexpressed in many cancers and promoted tumor cell proliferation through enhancing β-catenin accumulation and inhibiting Bcl-xl-mediated cell apoptosis[11]. Although these studies have demonstrated the importance of USP14 in cell physiology and diseases, the question of whether USP14 participates in other biological events, especially in energy metabolism, remains largely unexplored. Moreover, the global substrates of USP14 are still to be elucidated, representing a major bottleneck toward the functional characterization of USP14 and understanding of the complexity of proteasome-associated deubiquitination events[12,13].

Non-alcoholic fatty liver disease (NAFLD) is a major type of metabolic disorders, which has become a severe public health problem due to its high association with metabolic syndrome and progression of liver diseases, including non-alcoholic steatohepatitis, liver fibrosis, cirrhosis, and eventually hepatocellular carcinoma[14–16]. Aberrant triglycerides (TGs) accumulation in the liver promoted by obesity, is a hallmark feature of NAFLD[14,15]. Hepatic TG content is tightly regulated by de novo lipogenesis (DNL), fatty acid uptake, fatty acid oxidation and very low density lipoprotein (VLDL) export[17]. Steatosis develops when the amount of TG input is greater than the amount of TG output[17]. For example, the contribution of DNL to total hepatic TG contents in normal subjects is small and is much higher in obese patients[18,19]. Besides, hepatic overexpression of SREBP-1c or ChREBP, two master regulators of DNL, can result in hepatosteatosis[20,21]. Consistently, increased hepatic expression of several genes involved in DNL were observed in obese human and rodents[22–24]. However, the molecular mechanisms by which DNL is enhanced in obesity remain poorly understood.

In this study, by carrying out mass spectrometry-based deep proteome, ubiquitinome, and interactome analysis, we systematically screened the USP14 substrates. This study not only identified new possible substrates of USP14, but also revealed the broad new cellular pathways that USP14 were highly associated with, especially in lipid and carbohydrate metabolism. Consistently, we found that USP14 is increased in obese livers. We further demonstrated that fatty acid synthase (FASN), the terminal enzyme in DNL, is a specific substrate of USP14. USP14-mediated deubiquitination and stabilization of FASN promotes TG accumulation, revealing a novel mechanism of hepatosteatosis pathogenesis.

## Results

**Upregulation of USP14 in livers of obese mice.** We previously carried out Affymetrix arrays using livers of mice fed a high-fat diet (HFD) or a normal chow diet (ND)[25,26], and by reanalysis of these data we found that the mRNA levels of several USP family members were significantly upregulated ($p < 0.05$ by Student's t-test) in HFD mouse livers (Supplementary Fig. 1). Among these USP members, we found that the expression of USP14 was increased in HFD-fed mice (Supplementary Fig. 1). The upregulation of USP14 mRNA level was further confirmed by quantitative real-time PCR (Fig. 1a). In agreement, USP14 protein expression was increased in the livers of mice subjected to HFD (Fig. 1b).

Next, livers from leptin receptor-deficient mice (db/db) and NAFLD patients were examined. As a result, USP14 expression was higher in the livers of db/db mice and NAFLD patients, compared with the corresponding non-steatotic controls (Fig. 1c–f). Importantly, mRNA levels of USP14 correlated well with hepatic TG content (Fig. 1g). Together, our results demonstrate that upregulation of USP14 is a conserved feature of hepatosteatosis, suggesting that it may have an important role in the progression of NAFLD.

**Quantitative proteomic profiling of USP14 regulated proteins.** To system-wide identify the degradation substrates of USP14, we carried out mass spectrometry-based stable isotope labeling with amino acids in cell culture (SILAC) quantitative proteomics analysis (Fig. 2a). First, we chose HeLa cells as a cell line model, which contains relatively high level of endogenous USP14 and successfully constructed USP14 knockdown (KD) cells by specific shRNA transfection. Then, the proteome of USP14 KD cells was labeled with "heavy" ($^{13}C_6$-Lys and $^{13}C_6^{15}N_4$-Arg) amino acids, whereas proteome of control cells was labeled with "light" ($^{12}C_6$-Lys and $^{12}C_6^{13}N_4$-Arg) amino acids in cell culture. An equal amount of cell lysates extracted from the "light" and "heavy" cells was mixed, digested, fractionated, and analyzed by MS for proteome quantification. We identified 7647 protein groups in four biological replicates including a pair of reverse labeled cells (Fig. 2b). We defined significantly different ($p < 0.05$ by Student's t-test) proteins and used a criterion of 1.2-fold change or greater between these two groups as differential protein candidates. Subsequently, 108 downregulated proteins and 50 upregulated proteins in the USP14 KD group were identified (Supplementary Data 1).

**Characterization of ubiquitinome in response to USP14 KD.** To further characterize the deubiquitination substrates of USP14, we employed an affinity-based ubiquitinated peptide enrichment approach to systematically quantify the change of ubiquitinome in USP14 KD cells. As lysine residues modified with di-glycine remnant (K-ε-GG) will be derived from ubiquitinated proteins after tryptic digestion, we carried out affinity capture of ubiquitin-modified peptides using anti-di-glycine remnant pan antibody in a SILAC-labeled mixture of USP14 KD and control HeLa cells. The enriched K-ε-GG peptides were then subjected to nano-HPLC-MS/MS analysis for ubiquitinome quantification. In this experiment, we identified 15,241 ubiquitination sites, and 6562 sites were quantifiable (Fig. 2c, Supplementary Data 2). With a criterion of ≥1.2-fold change between these two groups, we finally obtained 392 significant upregulated ubiquitin sites ($p < 0.05$ by Student's t-test) (Fig. 2c, Supplementary Data 2), which

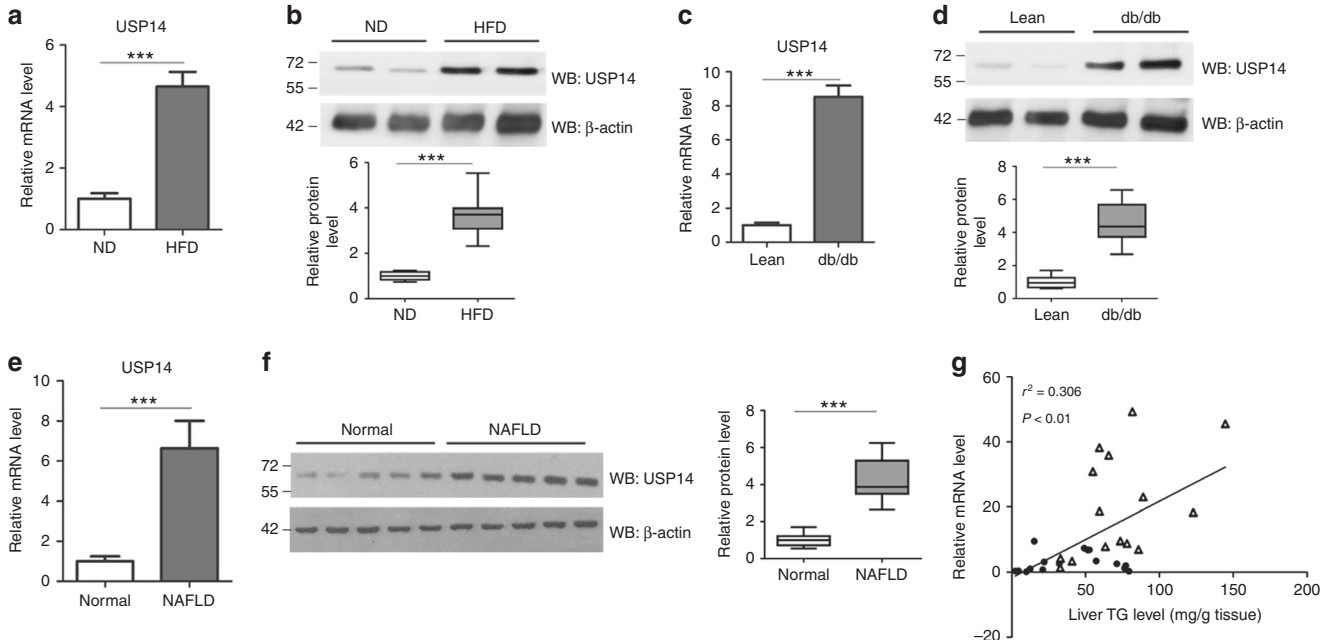

**Fig. 1** Upregulation of USP14 in livers from obese mice. **a**–**d** Hepatic mRNA and protein levels of USP14 in HFD obese mice **a**, **b** and leptin receptor-deficient *db/db* mice **c**, **d**. ($n = 8$–9 for each group). A representative western blot was shown and the quantification plot was based on scanning densitometry analysis using the Image J software (v 1.8.0). ***$p < 0.001$ versus indicated groups (Student's *t*-test). The top and bottom whiskers in **b**, **d** are the largest and smallest value, bounds of box show the 25th or 75th percentiles, and center line corresponds to the median. **e**, **f** Relative mRNA and representative protein expression of USP14 in livers from NAFLD patients ($n = 15$) and normal subjects ($n = 16$). A representative western blot was shown with the quantification plot based on scanning densitometry analysis. ***$p < 0.001$ versus indicated groups (Student's *t*-test). **g** Pearson R- and P-value for normalized USP14 mRNA levels versus TG content in human livers ($n = 31$). Data are presented as mean ± SEM. ***$p < 0.001$

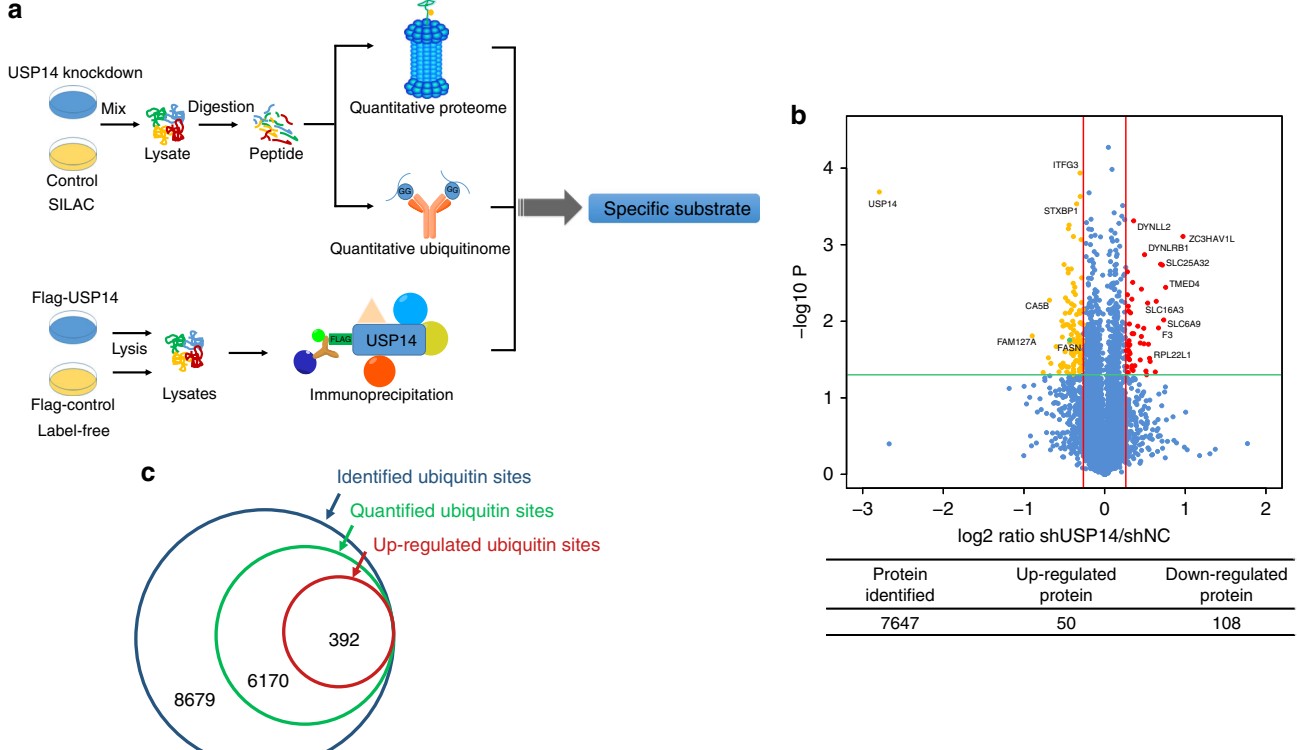

**Fig. 2** Quantitative proteomic profiling for USP14 regulated proteins and interaction components. **a** The diagram shows the experimental process for the quantitative proteomics, ubiquitinome, protein–protein interaction analysis of USP14. **b** Volcano plot of the protein abundance changes in response to USP14 KD. Average protein expression ratio of 4 replicates (log 2 transformed) between USP14 knockdown and wild-type HeLa cells was plotted against p-value by t-test (−log 10 transformed). Cutoff of $p = 0.05$ and 1.2-fold change were marked by green and red lines, respectively. The figure below shows the total number of proteins identified as well as the number of up- and downregulated proteins. The detailed information on protein groups is listed in Supplementary Data 1. **c** Number of identified ubiquitin sites, quantified ubiquitin sites, and upregulated ubiquitin sites

was likely to be regulated by USP14. To check whether there was any particular amino acid preference adjacent to the ubiquitinated sites, we analyzed the flanking sequences of upregulated ubiquitin sites using iceLogo software[27]. The result showed that glutamine and glutamic acid were notably over-represented close to the ubiquitin sites, whereas proline and leucine were less preferred near the ubiquitin sites (Supplementary Fig. 2).

**Annotation of USP14 regulated proteome and ubiquitinome**. To gain insight into the possible biological functions of USP14, we subjected the down- and up-regulated proteins in USP14 KD cells to bioinformatics enrichment analysis with gene ontology (GO) and Kyoto Encyclopedia of Genes and Genomes (KEGG) databases by the ClueGO tool[28]. In addition to the enrichment of the known USP14 participated pathways, such as proteolysis, apoptosis and NF-κB pathways (Fig. 3a, Supplementary Data 3), our analysis revealed that USP14 could also plays a role in multiple previously less well-appreciated cellular functions, such as fatty acid and amino acid metabolisms, epidermal growth factor receptor (EGFR), peroxisome proliferator-activated receptor (PPAR), and p53 signaling pathways (Fig. 3a, Supplementary Data 3). Notably, protein expression of the three key enzymes in the fatty acid synthesis pathway, ACLY, ACACA, FASN, and seven enzymes in the fatty acid degradation pathway were downregulated, which further indicated a possible role of USP14 in fatty acid metabolism (Fig. 3c). Consistently, protein interaction network analysis of the USP14 regulated proteins using the STRING Database[29] also identified fatty acid metabolism as one of the highly interconnected clusters (Supplementary Fig. 3a and Supplementary Data 3).

Consistently, bioinformatics analysis of upregulated ubiquitinated proteins in response to USP14 KD showed that they were closely related to the processes of fatty acid and carbohydrate metabolism (Fig. 3b, Supplementary Data 4). Notably, we found that ubiquitin levels at several lysine residues of the rate limiting enzymes in fatty acid metabolism and the tricarboxylic acid (TCA) cycle, including ACSL4 (Lys182, Lys546, Lys552), FH (Lys51), and FASN (Lys41, Lys70, Lys1072, Lys1142, Lys1151, Lys1158, Lys1866, Lys1878, Lys1993, Lys2449), were upregulated (Fig. 3c, Supplementary Fig. 3b). In addition to energy metabolism, the upregulated ubiquitinated protein were also enriched in chromatin functions, such as chromosome organization and chromatin DNA binding, and previously less well-recognized signaling events, such as FGFR, EGFR, VEGFR, and PI3K-Akt pathways, suggesting USP14 could be important in cellular epigenetics and multiple cellular signaling regulation (Fig. 3b, Supplementary Data 4). Together, the proteome and ubiquitinome analysis suggested multiple new roles of USP14 in cellular physiology, especially in lipid and carbohydrate metabolism.

**Identification of the potential direct substrates of USP14**. To further identify the direct substrates of USP14 that are involved in hepatosteatosis, we carried out immunoprecipitation assays and MS analysis (IP-MS) to investigate the interacting partners of USP14, as a substrate will physically interact with USP14 (Fig. 4a). We chose HEK293T cells, which is one of the most widely used and easily manipulated cells for transfection, to stably overexpress retrovirus-mediated Flag-tagged USP14 (Flag-USP14) or Flag-Control. The lysate of HEK293T cells with Flag-USP14 or Flag-Control was precipitated by anti-Flag M2 resin.

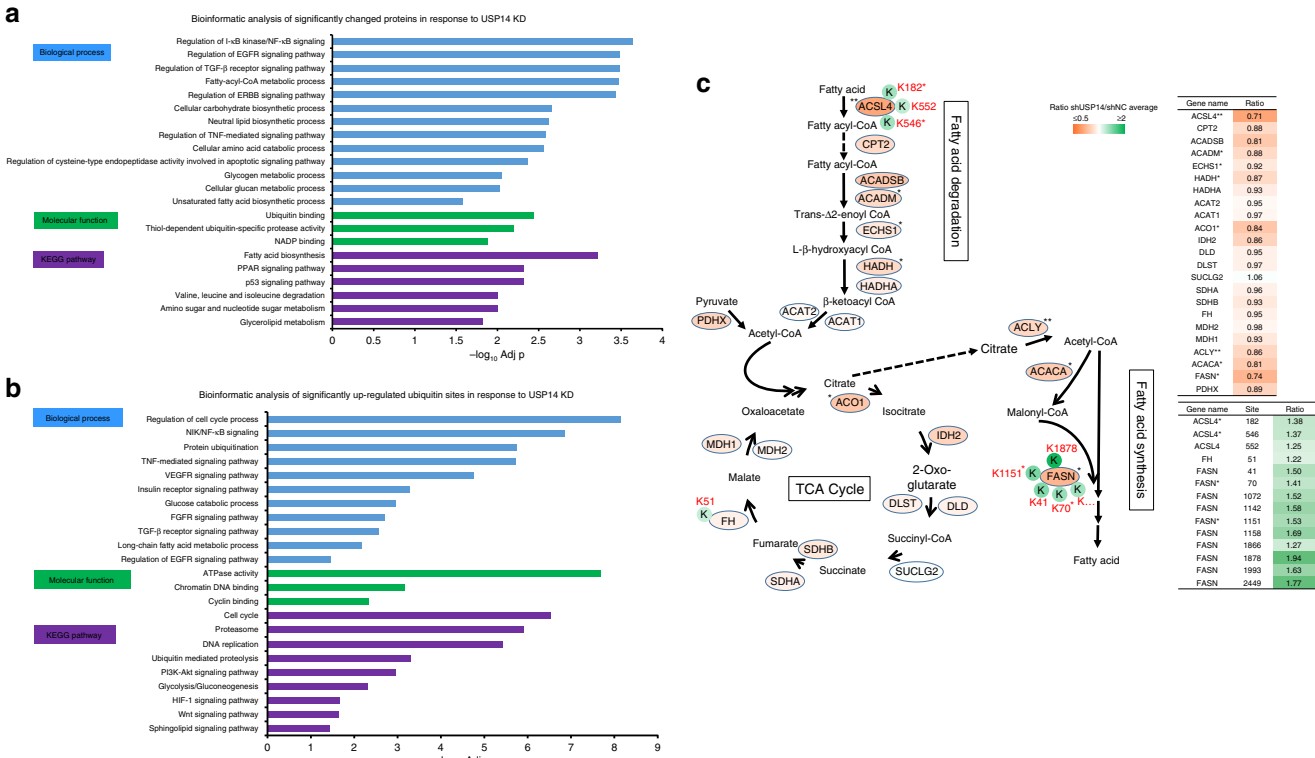

**Fig. 3** Functional annotation of altered proteome and ubiquitinome in response to USP14 KD. **a** GO analysis of the significantly changed proteins identified in SILAC process for biological process, molecular function, and KEGG pathway. **b** GO analysis of the significantly upregulated ubiquitin sites for biological process, molecular function, and KEGG pathway. **c** Fatty acid and TCA cycle pathway change in response to USP14 KD. Most key enzymes were significantly downregulated in metabolic processes, such as fatty acid degradation, fatty acid synthesis, and TCA cycle. Deficiency of USP14 leads to the decrease of fatty acid and lessens fatty acid degradation. *p < 0.05; **p < 0.01(Student's t-test)

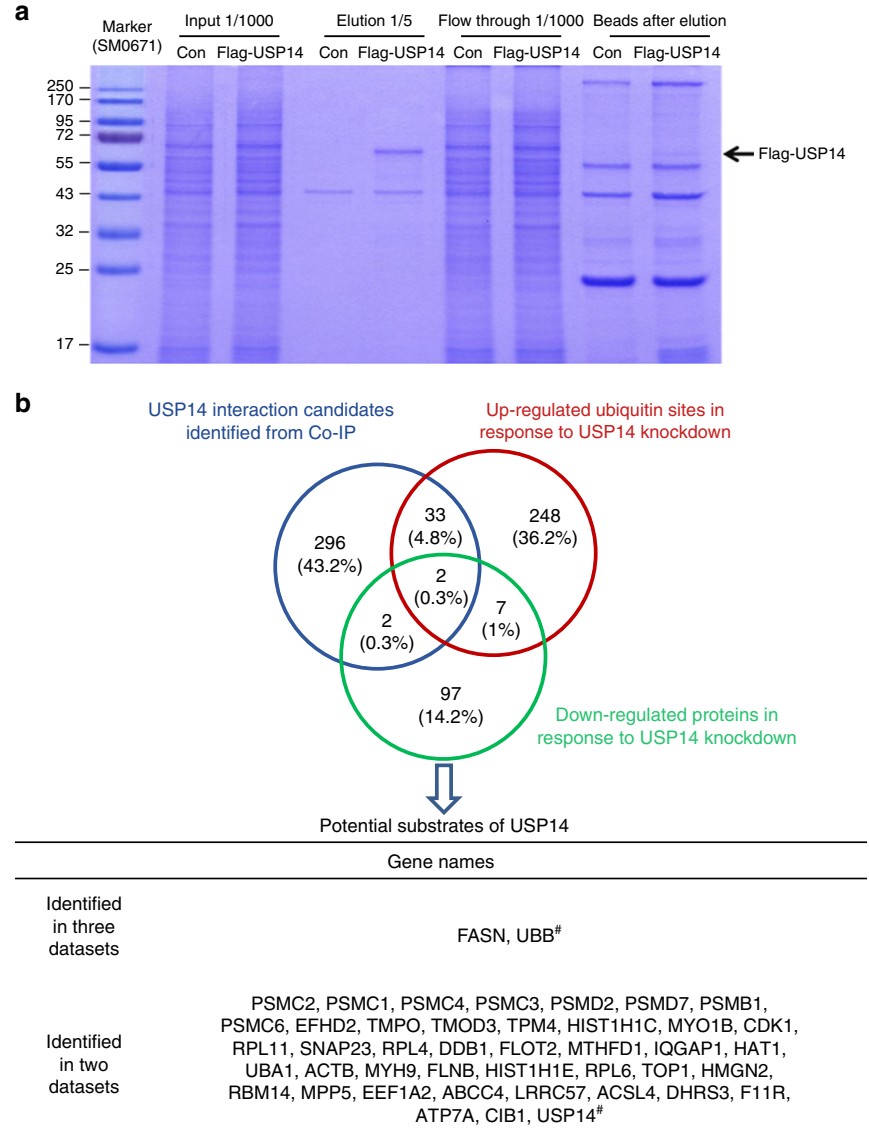

**Fig. 4** Purification of USP14 interacting proteins and identification of potential USP14 substrates. **a** The immunoprecipitation of Flag-USP14 interacting protein was visualized by Coomassie blue staining. **b** Venn diagram showing the number of downregulated proteins identified in response to USP14 KD (green), USP14 interaction candidates identified in Co-IP process (blue), upregulated ubiquitin sites identified in response to USP14 KD (red) and overlapping proteins in the datasets. The chart below shows potential substrates of the overlapped proteins

The exogenous expression of USP14 in pull-down elution was validated by SDS-PAGE (Fig. 4a). Then, the pull-down samples were digested with trypsin and analyzed by MS. We defined proteins with a fivefold change of the Mascot embedded label free quantitation emPAI (exponentially modified protein abundance index) score in at least two of the three biological replicates as potential substrates of USP14. Finally, we obtained 333 USP14 potential interacting proteins (Supplementary Data 5). Consistently, our result recaptured 21 of the 25 previously reported interacting partners of USP14[30], which were mainly proteasome regulatory subunit components.

To determine the direct substrates of USP14, we compared the downregulated proteins, upregulated ubiquitinated sites, and interacting proteins from the immunoprecipitation experiment. Ideally, a true substrate of USP14 will be identified from all three orthogonal independent experiments (i.e., proteome, ubiquitinome, and interactome). However, due to the technological limitations of each method, we considered that a protein identified from two of the three datasets could be a potential substrate of USP14 (see discussion for details). Using this criterion, we identified 42 potential substrates of USP14 (Fig. 4b, Supplementary Data 6). Interestingly, fatty acid synthase (FASN) were identified from all the three datasets in addition to the free ubiquitin (UBB). To further validate the IP-MS results, we carried out Western blot analysis of two proteins, FASN and CKB. The result showed that protein contents of FASN were significantly reduced in the USP14 KD cells compared with control cells, whereas CKB (identified only in one IP replicate) showed mild decrease (Supplementary Fig. 4). Therefore, FASN was considered as a possible USP14 candidate, whereas CKB was not included in our potential USP14 substrates in Fig. 4b according to our criteria.

**Verification of FASN as a USP14 target in endogenous cells**. As FASN meets all the biochemical characteristics of a direct substrate required for USP14 in our all MS experiments and given

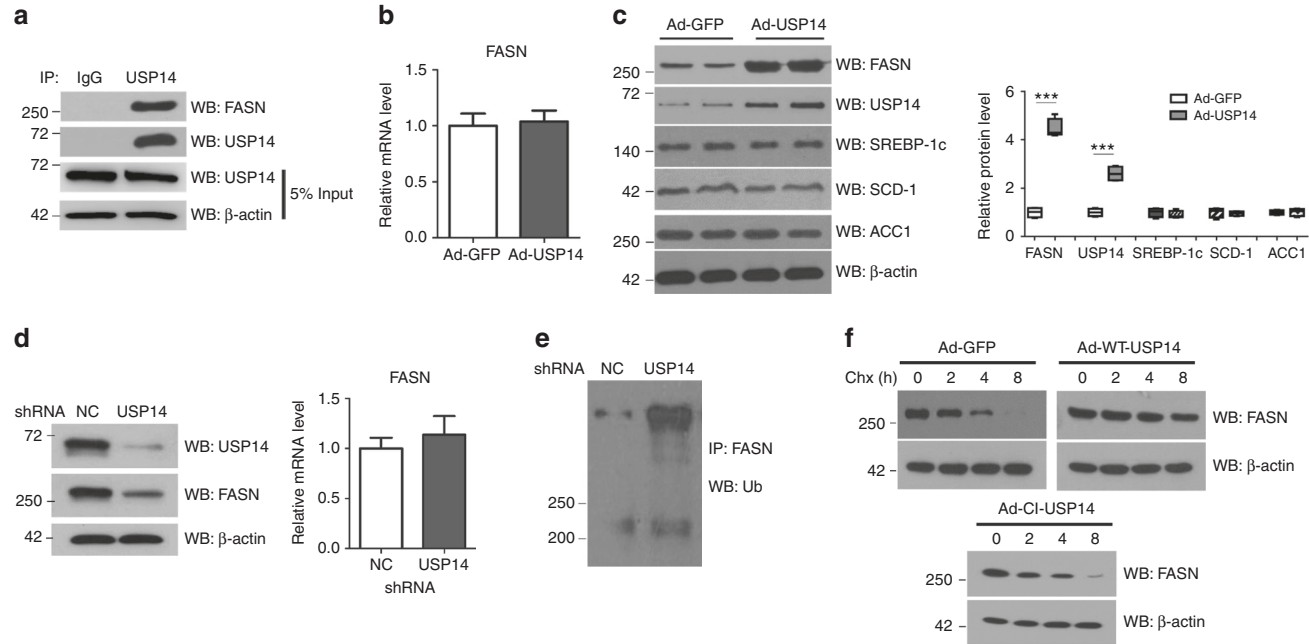

**Fig. 5** Identification of FASN as a direct target of USP14. **a** The lysate from mouse primary hepatocytes (MPH) were incubated with either anti-USP14 antibody or normal rabbit IgG, and the immunocomplexes were probed with the indicated antibodies. **b** Relative mRNA levels of FASN in MPH overexpressing USP14 or GFP for 48 h. **c** Representative protein levels of SREBP-1c, SCD-1 and ACC1 in MPH were shown and with the quantification plot based on scanning densitometry analysis (n = 4 for each group). The two lanes are two representative biological replicates. Each lane represents the tissue protein sample from one mouse. ***p < 0.001 versus indicated groups (Student's t-test). **d** MPH were transfected with shRNA targeting USP14 or negative control (NC) for 48 h. The protein levels of FASN, USP14 and β-actin were detected by western blot with the indicated antibodies and relative mRNA levels of FASN were determined by quantitative PCR. **e** MPH with or without USP14 knockdown were treated with 25 μg/mL cycloheximide (CHX) for the indicated time. The whole cell lysate was detected by western blot with the indicated antibodies. **f** MPH with or without USP14 overexpression were treated with 25 μg/mL cycloheximide (CHX) for the indicated time. The whole cell lysate was detected by western blot. Data are presented as mean ± SEM

the critical role of FASN in the regulation of fatty acid metabolism, we reasoned FASN as a major USP14 substrate candidate in hepatosteatosis for further investigation in endogenous cells. In mouse primary hepatocytes (MPH), FASN was detected in the anti-USP14 but not immunoglobulin G (IgG) immunoprecipitates from the cell lysate, suggesting the interaction of endogenous USP14 with FASN (Fig. 5a). Adenovirus-mediacted overexpression of USP14 resulted in increased FASN protein but not its mRNA expression levels in MPH (Fig. 5b), suggesting that USP14 regulates FASN expression at the post-translational level. Besides, protein abundance of other lipogenic genes, including SREBP-1c, SCD-1 and ACC1, were not affected by USP14 overexpression (Fig. 5c), suggesting a specific role of USP14 in FASN regulation. In contrast, knockdown of endogenous USP14 led to a decrease in FASN protein contents with unaffected mRNA expression (Fig. 5d), due to enhanced polyubiquitination of FASN (Fig. 5e). Next, we explored the half-life of FASN in MPH treated with the protein synthesis inhibitor cycloheximide (CHX). Co-expression of Flag-tagged wild-type (WT) USP14, but not catalytically inactive (CI) USP14[31] or GFP, significantly increased the FASN half-life (Fig. 5f), further supporting the notion that USP14 could stabilize FASN protein.

**USP14 overexpression promotes hepatosteatosis in mice.** The results that USP14 can regulate FASN protein stability prompted us to further investigate the role of USP14 in hepatic TG metabolism. Toward this goal, adenoviruses containing WT-USP14, CI-USP14[31], and GFP were administrated into C57BL/6 mice by tail vein injection, respectively. Our adenovirus did not induce liver inflammation and injury, compared with saline. As shown in

the Supplementary Fig. 5a, b, treatment of C57BL/6 mice with adenoviruses did not affect plasma ALT, AST levels and expression of pro-inflammatory cytokines, compared to saline treatment. As shown in Fig. 6a, USP14 was increased in the livers after injection of adenovirus. In agreement, protein levels of FASN were increased in mice with WT-USP14 overexpression, compared with the other two groups (Fig. 6a). In contrast, mRNA expression remained unaffected in all the three groups (Fig. 6a). Our results also showed that liver weights and TG contents were significantly elevated in WT-USP14-overexpressed mouse livers (Fig. 6b–d). Plasma TG levels were also increased (Fig. 6e). Due to a close association between hepatosteatosis and systemic insulin resistance[32,33], we observed a glucose intolerance and reduced insulin sensitivity in mice overexpressing USP14, as revealed by glucose and insulin tolerance tests (Supplementary Fig. 5c, d). In contrast, body weight, food intake, and body fat remained unaffected in USP14-overexpressed mice (Supplementary Fig. 5e–g).

To determine whether the increase of TG retention by USP14 overexpression relies on its regulation of FASN, we carried out adenoviral shRNA-mediated FASN knockdown combined with USP14 overexpression in C57BL/6 mice (Fig. 6f). As a result, knockdown of endogenous FASN expression largely blocked the roles of USP14 overexpression (Fig. 6g, h, Supplementary Fig. 5h, i), suggesting that the role of USP14 in the regulation of hepatic TG metabolism is, at least in part, dependent on upregulation of FASN.

Furthermore, it has been well-established that insulin plays a critical role in the regulation of hepatic lipogenesis and FASN expression. To determine whether the role of USP14 is dependent on insulin, C57BL/6 were treated with streptozotocin (STZ), which induced insulin deficiency due to the selective pancreatic

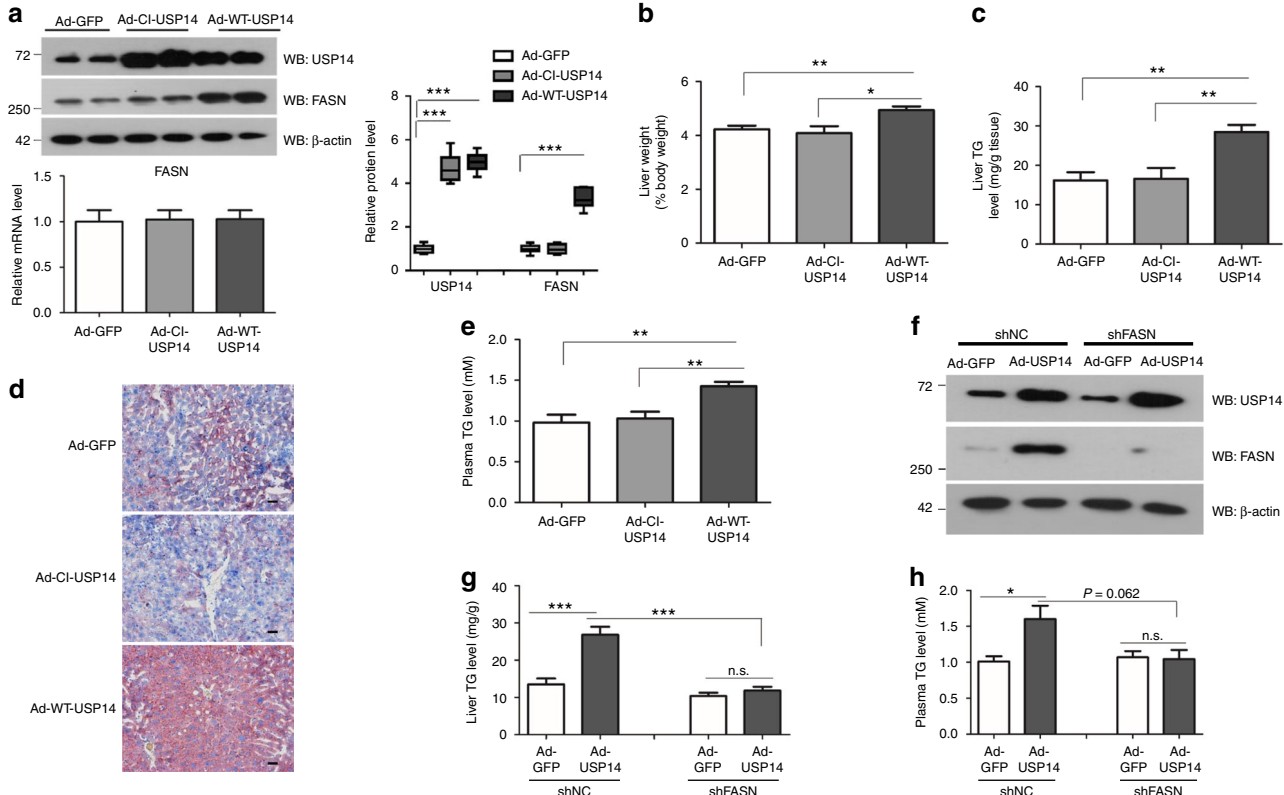

**Fig. 6** Overexpression of USP14 promotes hepatosteatosis. **a** Overexpression of hepatic USP14 in C57BL/6 mice infected with adenoviral wild-type (WT) USP14, catalytic inactive (CI) USP14, or green fluorescent protein (GFP) by western blot (up) and quantitative real-time PCR (bottom) ($n = 6$ for each group). The two lanes are two representative biological replicates. Each lane represents the tissue protein sample from one mouse. A representative western blot was shown and with the quantification plot based on scanning densitometry analysis (right). ***$p < 0.001$ versus indicated groups (Student's $t$-test). **b**, **c** Liver weight **b**, hepatic TG contents **c** in three groups of mice. **d** Representative hepatic histology by Oil Red O staining from mice overexpressing GFP, CI-USP14, or WT-USP14, scale bar 100 μm. **e** Plasma TG levels in three groups of mice. **f** Representative protein levels of USP14 in mice administrated with adenovirus as indicated. **g**, **h** Hepatic and plasma TG contents in four groups of mice. After a 16-h fast, mice were killed at day 10. Two-way ANOVA was used for statistical analysis. Data are presented as mean ± SEM. *$p < 0.05$; **$p < 0.01$; ***$p < 0.001$

beta-cell toxicity. Fourteen days later, mice were administrated with adenovirus containing USP14 or GFP via tail vein injection. As a result, overexpression of USP14 increased liver triglyceride contents and FASN protein levels (Supplementary Fig. 6a, b). In addition, it has recently been reported that USP14 negatively regulates autophagy[34], which plays an important role in the regulation of hepatic TG metabolism. To clarify this issue, mice were administrated with adenovirus containing USP14 or GFP. Five days later, mice were treated with 3-MA, an autophagy inhibitor. As a result, inhibition of autophagy did not reverse the TG accumulation in Ad-USP14 mice liver (Supplementary Fig. 6c). Therefore, our results indicate that the role of USP14 may be independent on insulin or autophagy.

**Ablation of USP14 improves hepatosteatosis in obese mice**. To further investigate the role of USP14 in obesity-related hepatosteatosis, two approaches to block USP14 function were employed. First, *db/db* mice were daily administrated with two doses of IU1, a specific small molecule inhibitor of USP14[34,35]. As a result, IU1 treatment led to a reduced TG contents and FASN protein levels in *db/db* mice in a dose-dependent manner (Supplementary Fig. 7a–c). Additionally, to rule out the non-specific effect of IU1, two adenoviral shRNAs (shUSP14-1, shUSP14-2) or negative control (NC) was adopted to silence hepatic USP14 expression in *db/db* mice. Treatment of db/db mice with adenoviral shRNAs did not affect plasma ALT, AST levels and expression of pro-

inflammatory cytokines, compared to saline treatment (Supplementary Fig. 8a, b). Our results showed that protein, but not mRNA levels of FASN, were reduced in Ad-shUSP14 mice as compared to Ad-shNC mice (Fig. 7a). As expected, the liver weight was reduced in *db/db* mice treated with Ad-shUSP14 (Fig. 7b). Hepatosteatosis was greatly alleviated and TG contents were reduced in Ad-shUSP14-infected *db/db* mice (Fig. 7c, d). Consistently, abrogation of hepatic USP14 resulted in a reduction in plasma TG levels (Fig. 7e).

Knockdown of USP14 expression also improved hyperglycemia, hyperinsulinemia, and insulin resistance in *db/db* mice (Fig. 7f–h). Phosphorylated AKT (Serine 473) was enhanced while gluconeogenic genes (PEPCK and G6Pase) were downregulated in shUSP14-infected db/db mice (Supplementary Fig. 8c, d). In contrast, body weight, food intake, and total body fat contents were comparable in the two groups (Supplementary Fig. 8e–g).

**Regulation of hepatic USP14 by LXR activation in obesity**. The results above have shown that USP14 was upregulated in the obese livers, and overexpression of USP14 promoted obesity-induced hepatosteatosis. Finally, we sought to identify the signaling pathway that may regulate USP14 expression in the liver. It has been shown that nuclear receptors, including liver X receptor (LXR), fanesoid X receptor (FXR), glucocorticoid receptor (GR) and peroxisome proliferator-activated receptor α

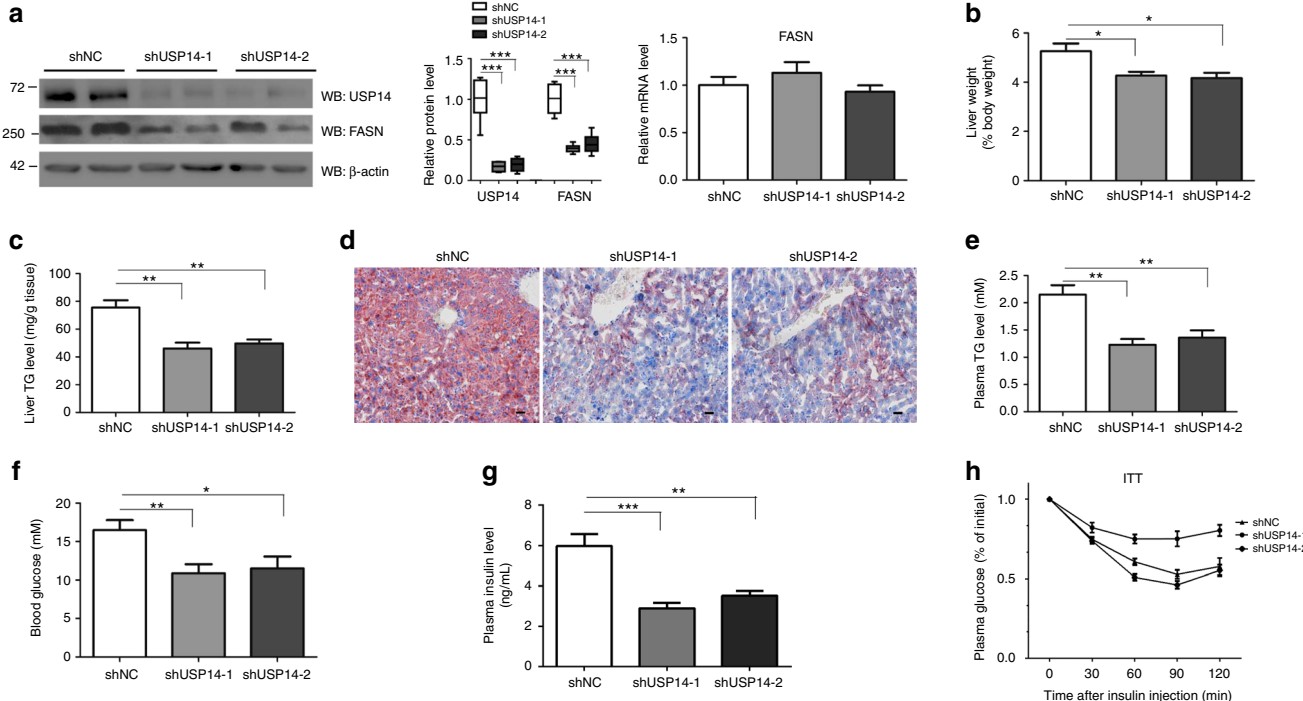

**Fig. 7** Knockdown of USP14 improves hepatosteatosis in *db/db* mice. **a** Hepatic USP14 protein (left) and mRNA (right) levels in *db/db* mice. Mice were administered with adenoviral shRNAs targeting USP14 or NC via tail vain injection. (*n* = 8 for each group). A representative western blot was shown with the quantification plot based on scanning densitometry analysis. The two lanes are two representative biological replicates. Each lane represents the tissue protein sample from one mouse. \*\*\**p* < 0.001 versus indicated groups (Student's *t*-test). **b**, **c** Liver weight **b**, hepatic TG contents **c** in *db/db* mice infected with shNC and shUSP14. **d** Representative hepatic histology by Oil Red O staining from *db/db* mice, scale bar 100 μm. **e** Plasma TG levels in two groups of mice. **f**–**h** Blood glucose and insulin levels and insulin tolerance tests in *db/db* mice. After a 16-h fast, mice were killed at day 12. Data are presented as mean ± SEM. \**p* < 0.05; \*\**p* < 0.01; \*\*\**p* < 0.001

(PPARα) play crucial roles in the regulation of genes responsible for hepatic lipids metabolism[36–38]. Therefore, MPH were treated with agonists of these nuclear receptors. As a result, we found that USP14 expression was specifically induced by TO901317, a LXR agonist (Fig. 8a), whereas agonists of other nuclear receptors, including GW4064, dexamethasone and WY14643, did not affect its expression. Therefore, we speculate that USP14 might be a molecular target of LXR. To test it, we examined the promoter region of USP14 and found that a potential LXR-binding motif exists in the proximal promoter region of USP14 gene (Fig. 8b). Luciferase reporter assays further showed that LXR increased the transcriptional activity of this promoter, especially in the presence of TO901317. However, mutation of the motif largely attenuated the trans-activated roles of LXR (Fig. 8c). Chromatin immunoprecipitation (ChIP) assays further showed that TO901317 could facilitate the recruitment of LXR and RNA Pol II to the LXRE region of the *USP14* gene (Fig. 8d). Therefore, we demonstrate that LXR signaling may be an important mechanism leading to upregulation of USP14 in obese livers.

## Discussion

USP14 is a well-known negative regulator for proteasomal activity. Recent studies have shown that USP14 participates in cell signaling transduction via regulating the turnover of its specific substrates, which in turn have important functional consequences in neuroglial diseases and cancers. Despite these effects, the direct substrates and additional roles of USP14 remain largely unknown. Theoretically, a true proteolysis-associated substrate of USP14 will be degraded by USP14 KD through its elevated protein ubiquitination level, and will interact with USP14. To this

end, we carried out the first comprehensive proteome-wide quantitative analysis of USP14 regulated proteome, ubiquitinome and interactome.

However, each of the technologies employed in this study has its own limitations. Despite the rapid advancement, current MS-based proteome analysis still cannot achieve the complete coverage of the highly dynamic cellular proteome. Therefore, our quantitative proteome analysis is likely to miss low abundant proteins. Our ubiquitinome analysis employed a K-GG antibody enrichment approach of tryptic peptides. The enriched K-GG peptides could be bias toward antibody specificity. Our immunoprecipitation experiment is likely to capture only the high affinity interaction partners of USP14, and the weak interaction partners are unlikely to be missed. Due to these limitations, we therefore considered that if a potential substrate of USP14 could be overlapped from two of the three independent datasets.

Our proteome analysis identified 108 downregulated proteins in response to USP14 KD, which revealed the first landscape of USP14 participated cellular pathways and networks. Our results not only identified the well-recognized functions of USP14, such as proteolysis and apoptosis, but also disclosed its new potential roles in lipid and energy metabolism, suggesting a broad regulatory role of USP14 in various cellular functions. Consistently, our ubiquitinome analysis showed that USP14-associated deubiquitination plays critical role in a variety of cellular processes, especially in lipid metabolism, signaling transduction, and chromatin functions.

By comparison of proteome and ubiquitinome data, we found that 225 ubiquitinated proteins were upregulated by USP14 KD, but their protein expression did not show obvious change in our proteome data. From these proteins in which the ubiquitination

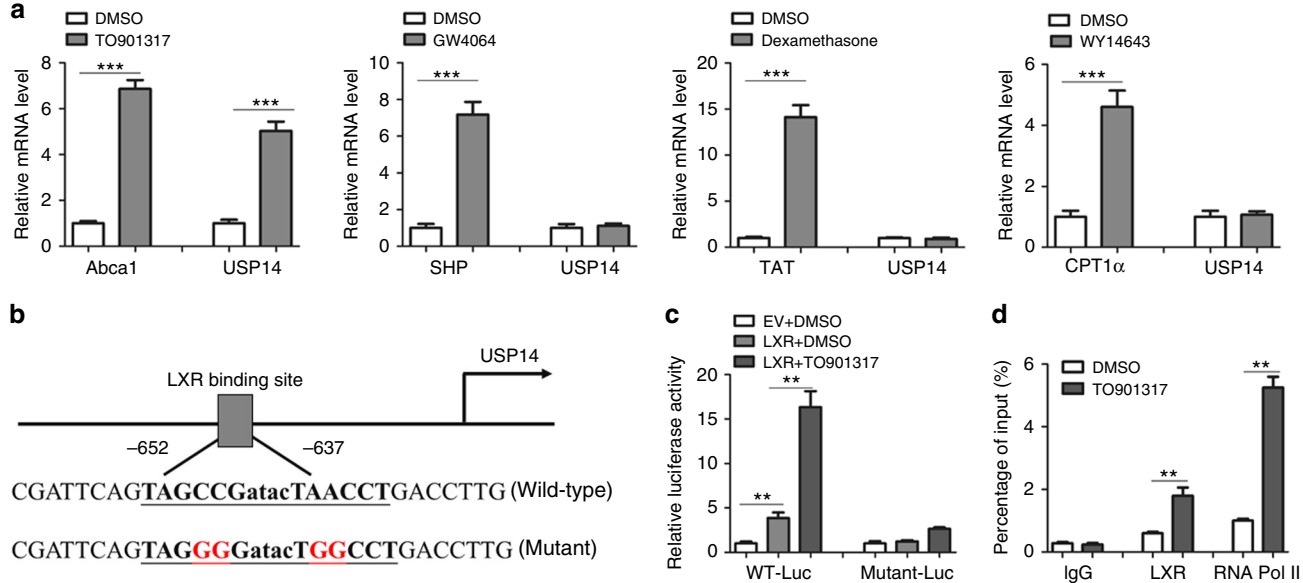

**Fig. 8** Regulation of USP14 expression by LXR. **a** Relative mRNA levels of USP14 in MPHs treated with TO901317, GW4064, dexamethasone, WY14643 or vehicle control (DMSO) for 16 h. Expression of Abca1, SHP, TAT, and CPT1 were used as positive controls for LXR, FXR, GR, and PPAR activation, respectively. **b** Proximal promoter region of mouse USP14 gene contains a potential binding site for LXR. **c** Luciferase reporter assays. HEK293T cells were transfected with luciferase reporter plasmids containing wild-type (WT-Luc) or mutant (Mut-Luc) binding site of LXR. Cells were treated with vehicle control (DMSO) or TO901317 for 12 h. **d** Chromatin immunoprecipitation assays showing representative LXR and RNA Pol II binding to the USP14 promoter in MPHs. Cells were treated with TO901317 or DMSO for 4 h and then subjected to ChIP assays. Data are presented as mean ± SEM. $**p < 0.01$; $***p < 0.001$ (Student's $t$-test)

level upregulated but expression-unchanged, we additionally found 29 proteins were identified from our interactome data from the immunoprecipitation experiment (Supplementary Data 6). Together, these ubiquitination level upregulated USP14 interacting proteins could also potentially be the non-proteolytic substrates of USP14, or proteolytic substrates that are compensated by other DUB enzymes, such as UCH37[5]. The non-proteolytic role of USP14 in cellular function remains to be further investigated.

Particularly, our results proposed a novel function of USP14 in the regulation of hepatic TG homeostasis. This was suggested by multiple lines of evidence. First, both the proteome and ubiquitinome analysis in response to USP14 KD suggested the role of USP14 in fatty acid metabolism. Second, our protein affinity purification identified FASN as a novel interacting protein of USP14. USP14 could enhance FASN stability by blocking its ubiquitination and degradation. Third, selective overexpression of hepatic USP14 resulted in massive TG accumulation in normal mice, whereas the hepatic silence of USP14 significantly attenuated hepatosteatosis in db/db mice. Forth, our results demonstrated that FASN could be a direct target of USP14 to exert its function in hepatic DNL. Thus, we propose that USP14 could be an important mediator for obesity-related hepatosteatosis. Although, we showed the upstream LXR signaling was associated with USP14 expression in the liver, additional mechanisms that underlie its upregulation in obese livers need to be uncovered in future studies. For example, it has been shown that USP14 expression is regulated by several microRNAs (miRNAs), including miR-4782-3p and miR-124[39,40]. Whether these miRNAs are involved in the regulation of USP14 in obese livers remains to be determined. Another limitation in our study is the lack of liver-specific USP14 knockout mice. The exact role of USP14 in fatty liver could be better clarified if this animal model was used in the future.

FASN, the major enzyme converting acetyl-CoA into fatty acids, plays a central role in the hepatic de novo lipogenesis and TG homeostasis. It has been well-established that the expression level of FASN is increased in the livers of obese rodents and humans[41]. At the transcription level, FASN is regulated by SREBP-1c, ChREBP, and USF-1 in hepatocytes upon treatment with nutrients, such as glucose and fatty acids[42]. Especially, FASN expression is also induced by LXR ligands. LXRs activate FASN expression through direct interaction with the FASN promoter as well as through upregulation of SREBP-1c[42]. Therefore, we conclude that LXR could regulate FASN expression by transcriptional and post-transcriptional level.

Besides, Yu et al. reported that the Src-homology 2 (SH2) domain-containing tyrosine phosphatase (Shp2) acts as an adaptor molecule connecting ubiquitin E3 ligase COP1 with FASN, thereby regulating FASN ubiquitination and degradation[43]. Moreover, USP2a has been shown to interact with and stabilize FASN in prostate cancer cells[44]. In addition, expression and activity of liver FASN correlate with O-GlcNAcylation contents in obese mice both in a transcription-dependent and -independent manner[45]. Inhibition of its O-GlcNAc modification increases the interaction between FASN and USP2a in vivo and ex vivo, providing an insight into the control of FASN expression by O-GlcNAcylation[45]. Thus, both these studies and our findings indicate that characterization of novel regulatory molecules for FASN expression might reveal novel mechanisms into disease pathogenesis. It also raised the question that whether UPS14, or other liver-enriched deubiquitinase, could also bind to FASN and stabilize the protein via a crosstalk between ubiquitination and a nutrient-dependent post-translational modification, such as O-GlcNAcylation. Further studies are still needed to address these questions. On the other hand, upregulation of FASN was also observed in many types of human cancers, which is required for tumor growth and metastasis[45,46]. Therefore, it will be interesting to explore whether USP14 could regulate FASN protein stability in cancer cells to promote tumorigenesis.

In conclusion, the results of our present study demonstrate that USP14 plays an important role in hepatosteatosis through FASN stabilization and represents a therapeutic target in fatty liver and related metabolic disorders.

## Methods

**Animal experiments and adenovirus preparation**. Male C57BL/6, and *db/db* mice aged 8 weeks were purchased from the Shanghai Laboratory Animal Company (SLAC) and Nanjing Biomedical Research Institute of Nanjing University, respectively. All mice were housed at 21 °C ± 1 °C with humidity of 55 ± 10% and a 12-h light/12-h dark cycle. HFD-induced obese mice were maintained with free access to high-fat chow (D12492; Research Diets) for 12 weeks. Adenovirus-expressing murine USP14 was constructed by Invitrogen (Shanghai, China) with a full-length USP14 complementary DNA (cDNA) coding sequence. Overexpression of hepatic USP14 was achieved by means of tail vein injection of Ad-USP14 ($2 \times 10^9$ plaque-forming units) in normal C57BL/6 mice. To determine whether the role of USP14 is dependent on insulin, C57BL/6 were treated with streptozotocin (STZ, 240 mg/kg). Fourteen days later, mice (blood glucose levels exceeding 300 mg/dL) were administered with adenovirus containing USP14 or GFP via tail vein injection. Ten days later, mice were killed for analysis. To inhibit autophagy, some mice overexpressing USP14 were administered with 3-MA (30 mg/kg/day, i.p.) for 5 days. To silence USP14 expression in *db/db* mice, two adenoviruses expressing USP14 shRNA ($1 \times 10^9$ plaque-forming units) were generated using pAD_-BLOCK_IT_DEST vectors (Invitrogen, Grand Island, New York, USA). All viruses were purified by the cesium chloride method and dialyzed in PBS containing 10% glycerol prior to animal injection. The animal protocol was reviewed and approved by the Animal Care Committee of Zhongshan Hospital, and conducted in accordance with all relevant ethical regulations.

**Human samples**. For analysis of hepatic UPS14 expression and TG contents, liver biopsy was performed in those subjects who donated their partial livers for liver transplantation. The subjects were screened with physical examination and type B ultrasonography. The clinical characteristics of study subjects are shown in Supplementary Data 7. The human study was approved by the Human Research Ethics Committee of Zhongshan Hospital and conducted in accordance with all relevant ethical regulations. Informed consent was obtained from each human participants.

**Glucose and insulin tolerance tests**. Glucose tolerance tests (GTTs) were performed by intraperitoneal injection of D-glucose (Sigma-Aldrich) at a dose of 2.0 mg/g body weight after a 16-h fast. For insulin tolerance tests (ITTs), mice were injected with regular human insulin (Eli Lilly and Company, Indianapolis, IN) at a dose of 0.75 U/kg body weight after a 6-h fast. Blood glucose was measured using a portable blood glucose meter (LifeScan; Johnson & Johnson, New Brunswick, NJ).

**Hepatic triglyceride measurement**. Liver tissues (weighing ∼100 mg) were collected and homogenized in chloroform/methanol (2:1 v/v) using a Polytron tissue grinder. The extracts were dried under nitrogen flow and resuspended in isopropanol. TG concentrations were measured using commercial kits (Sigma, St. Louis, MO, USA; Biovision, Milpitas, California, USA) according to the manufacturer's instructions.

**Cell culture and SILAC labeling**. All cells used in cell culture and SILAC experiments were bought from ATCC. The cells were regularly tested for mycoplasm contamination by Hoechst DNA staining and found to be negative. HEK293T WT cells and Flag-USP14 overexpressing 293T cells were cultured in Dulbecco's Modified Eagle Medium (DMEM) medium containing penicillin-streptomycin solution and 10% FBS. HeLa cells were cultured in DMEM with light lysine ($^{12}C_6^{14}N_2$-Lys) and light arginine ($^{12}C_6^{14}N_4$-Arg) containing penicillin-streptomycin solution and 1X GlutaMAX and 10% dialyzed FBS, USP14 KD Hela cells were cultured in Dulbecco's Modified Eagle Medium (DMEM) with heavy lysine ($^{13}C_6$-Lys) and heavy arginine ($^{13}C_6^{15}N_4$-Arg) containing penicillin-streptomycin solution and 1X GlutaMAX and 10% dialyzed FBS. All the cells were cultured in 37 °C with 5% $CO_2$. SILAC labeling cells were cultured more than six generates and the labeling efficiency was determined by mass spectrum before the proteomics study, the labeling efficiency of the cells cultured in heavy medium was over 97%. Four biological replicates (3 forward SILAC labeling and 1 SILAC reverse labeling) and three technical replicates were prepared for proteome analysis.

**Mouse primary hepatocyte isolation**. Mouse primary hepatocytes were isolated from C57BL/6 mice by collagenase perfusion and purified by centrifugation. Fresh hepatocytes were seeded in 6-well plates in attachment media (Science Cell, USA). The media were then replaced with DMEM (Gibco, USA) within 24 h.

**Sample preparation for mass spectrometry data analysis**. All samples were washed three times with ice-cold Dulbecco's PBS (Mediatech Inc., Manassas, VA), lysed in chilled lysis buffer (8.0 M urea in 0.1 M $NH_4HCO_3$ supplemented with 1× protease inhibitor cocktail (Calbiochem, Darmstadt, Germany), and incubated on ice for half an hour. After sonication, the debris was removed by centrifugation and the supernatant was collected. The protein concentration of the supernatant was determined by BCA assay. Extracted protein (200 μg) of each sample was reduced with 5 mM dithiothreitol (DTT) (Acros, Belgium) at 56 °C for 30 min. Then, the samples were incubated with 15 mM iodoacetamide (IAA) (Acros, Belgium) at room temperature in the darkness for 30 min and quenched by 15 mM DTT. Samples were digested by trypsin (an enzyme-to-substrate ratio of 1:50 (w/w)) at 37 °C overnight and then digested by trypsin (an enzyme-to-substrate ratio of 1:100 (w/w)) for additional 4 h.

**Ubiquitinated peptide enrichment**. Light-labeled HeLa cell lysate and heavy-labeled USP14 KD HeLa cell lysate were mixed equally, digested and separated. Five percent of the peptides were used for protein quantification; whereas, the remaining peptides were used for ubiquitin peptides enrichment. Each fraction was dissolved in NETN buffer (100 mM NaCl, 1 mM EDTA, 50 mM Tris-HCl, 0.5% NP-40, pH 8.0) and then incubated separately with pre-washed antibody beads (Cell Signaling Technology, Beverly, MA, USA and PTM Biolabs, Hangzhou, China) at 4 °C overnight with gently shaking. The bound peptides were eluted with 0.1% TFA and vacuum-dried followed by LC–MS/MS analysis. Two biological replicates and two technical replicates were carried out for ubiquitinome analysis.

**Immunoprecipitation (IP)**. Cells were lysed in lysis buffer for 20 min with gentle rocking at 4 °C. Lysates were cleared by centrifugation and then filtered through 0.45 μm spin filters (Millipore) to further remove cell debris, and the resulting material was subjected to IP with 50 μL anti-FLAG M2 affinity resin (Sigma) overnight at 4 °C. Resin-containing immune complexes was washed with ice-cold lysis buffer followed by Tris buffered saline (TBS) washes. Proteins were eluted with two-50 μL 150 μg/mL 3× Flag-peptide (Sigma) in TBS for 30 min, and the elution was pooled for a final volume of 100 μL. Proteins in each elution were precipitated with cold acetone. Three biological replicates were carried out.

**Nano-HPLC-MS/MS and mass spectrometry data analysis**. The precipitated protein was digested by trypsin (trypsin: protein = 1:50 (w/w)) at 37 °C for 16 h. The peptides were then analyzed by an EASY-nLC 1000 system connected to an Q Exactive mass spectrometer and Orbitrap Fusion mass spectrometer (Thermo Fisher Scientific, Waltham, MA, USA). The proteome data were acquired by Q Exactive with high resolution MS2 spectrum and the ubiquitinome and interactome data were acquired by Oribtrap Fusion with low resolution MS2 spectrum. Peptides were eluted from a homemade reverse-phase C18 column (75 μm ID, 3 μm particle size, Dikma Technologies Inc.) with a 70 min gradient of 7% to 80% buffer B (90% acetonitrile, 10% $H_2O$, 0.1% formic acid) at a flow rate of 300 nL/min. For the Q Exactive mass spectrometer, the automatic gain control (AGC) targets were 1.0e6 for full scan and 1.0e6 for MS/MS scan, respectively. Full MS spectra with an m/z range of 350–1300 were acquired with a resolution of 70,000 at $m/z = 200$ in profile mode. Fragmentation of the 16 most intense precursor ions occurred at the high-energy collision dissociation (HCD) collision cell with a normalized collision energy of 28%, and tandem MS were obtained with a resolution of 17,500 at m/z = 200. Dynamic exclusion duration was 60 s and ions with a single charge or more than six charges were excluded from tandem mass fragmentation. For the Orbitrap Fusion mass spectrometer, the automatic gain control (AGC) targets were 1.0e6 for full scan and 7e3 for MS/MS scan, respectively. Full MS spectra with an m/z range of 350–1300 were acquired with a resolution of 120,000 at $m/z = 200$ in profile mode. MS/MS acquisition was performed in top speed mode with 3 s cycle time occurred at the high-energy collision dissociation (HCD) collision cell with a normalized collision energy of 32%. Tandem MS were obtained in the ion trap. Dynamic exclusion duration was 60 s and ions with a single charge or more than six charges were excluded from tandem mass fragmentation. The MS/MS data were processed using Mascot software (version 2.3.0) and MaxQuant software (version 1.4.1.2). For database searching, trypsin was set as the specific enzyme and the maximum number of missed cleavages was fixed at 2, and all data were searched against UniProt Human database (88,817 sequences, release date: May 2013). The precursor mass tolerance was 10 ppm for Mascot software and 20 ppm for MaxQuant software; and the fragments mass tolerance was 20 ppm for the proteome data and 0.5 Da for the ubiquitome and interactome data. The modification searching parameters were set as follows: Carbamidomethyl (C) was set as fixed modification, and Acetyl (Protein N-term), GlyGly (K), Oxidation (M), Label: $^{13}C_6$-Lys and Label: $^{13}C_6^{15}N_4$-Arg as variable modifications. The proteins and peptides FDR were 1% for MaxQuant software. A Mascot peptide ion score of at least 20 and two peptides were used as cutoff for Co-IP protein identification. The Exponentially Modified Protein Abundance Index (emPAI) score from Mascot software was used for Co-IP interacting protein analysis. Proteins which could be identified in two of the four biological replicates were used for statistical analysis. Statistical differences were determined by two-tailed Student's *t*-test. *P*-value <0.05 and 1.2-fold-of-change was considered significant for the proteome and ubiquitome data. *P*-value <0.05 and fivefold-change of emPAI score was considered significant for the interactome data.

**Bioinformatic analysis**. The functional enrichment analysis was implemented using ClueGO (version 2.2.5)[28] or The Database for Annotation, Visualization and Integrated Discovery (DAVID) Bioinformatics Resources 6.7[47,48]. The GO FAT database from DAVID was selected for the enrichment. The whole-Homo-sapiens genome information was set as the background. A Benjamini $p$-value cutoff of 0.05 was used to control the FDR.

**Western blots**. Total protein was separated by 12% sodium dodecyl sulfate polyacrylamide gel electrophoresis (SDS-PAGE) and transferred onto poly-vinylidene fluoride membranes. Membranes were blocked in 5% non-fat milk in phosphate-buffered saline with Tween-20 (PBST) for 1 h before incubation with primary antibody diluted with 5% non-fat milk in PBST overnight at 4 °C. Membranes were washed with PBST five times and incubated with secondary antibody for 1 h. The signals of the proteins were visualized on an ImageQuant LAS 4000 system (GE Healthcare). The following primary antibodies were used: anti-USP14 antibody at 1:2000, (#11931, Cell Signaling Technology, Beverly, MA, USA), anti-FASN antibody (C20G5) at 1:1000, (#3180, Cell Signaling Technology, Beverly, MA, USA), anti-creatine kinase B type antibody at 1:1000, (ab125114, Abcam, Cambridge, UK), anti-SREBP-1c antibody at 1:2000, (sc-367, Santa cruz); anti-SCD-1 antibody at 1:1000, (2438, Cell Signaling Technology); anti-Akt anti-body at 1:2000, (9272, Cell Signaling Technology); anti-p-Akt antibody at 1:1000, (Ser473, 4060S, Cell Signaling Technology); anti-ACC1 antibody at 1:1000, (3662, Cell Signaling Technology), and anti-β-actin antibody at 1:10,000, (#3700, Cell Signaling Technology). All the films for WB analysis presented in this study with molecular weight markers are shown in Supplementary Fig. 9.

**RNA isolation and real-time PCR**. Total RNA was isolated from liver tissues using TRIzol (Invitrogen) according to the manufacturer's instructions. In order to quantify the transcripts of interest genes, quantitative real-time PCR was performed using a SYBR Green Premix Ex Taq (Takara) on Light Cycler480 (Roche, Switzerland). Primers were selected from PrimerBank (https://pga.mgh.harvard.edu/primerbank/) and listed as follows: USP14 (Forward: 5′-ATGCCACTC TACTCTGTTACAGT-3′; Reverse: 5′-AACACCATTGGAGGTTCATCAG-3′); FASN (Forward: 5′-GGAGGTGGTGATAGCCGGTAT-3′; Reverse: 5′-TGGGTA ATCCATAGAG CCCAG-3′); 36B4 (Forward: 5′-AGATTCGGGGATATGCTGTT GGC-3′; Reverse: 5′-TCGGGTCCTAGACCAGTGTTC-3′). Relative quantitation analysis of gene expression data was conducted according to the $2^{-\Delta\Delta Ct}$ method.

**Luciferase reporter and chromatin immunoprecipitation assays**. The promoter region of USP14 gene was amplified from the mouse genomic DNA and inserted into pGL4.15 vector (Promega, Madison, Wisconsin, USA). For the luciferase reporter assays, HEK293T cells were seeded in 24-well plates and transfected with the indicated plasmids using Lipofectamine 2000 (Invitrogen). Cells were collected 36 h after transfection. Luciferase activity was measured using the Dual Luciferase Reporter Assay System (Promega). A chromatin immunoprecipitation (ChIP) assay kit was used (Upstate, Billerica, Massachusetts, USA). In brief, MPH were fixed with formaldehyde. DNA was sheared to fragments at 200–1000 bp using sonications. Chromatin was incubated and precipitated with antibodies against LXR, RNA Poll II or IgG (Santa Cruz).

## Data availability

All mass spectrometry raw data have been deposited to the iProX Consortium, a full member of the ProteomeXchange consortium, with the dataset identifier (Project ID) "IPX0001138000".

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

## Acknowledgements

This work was supported by the National Key Research and Development Program of China (No. 2016YFC1304801), National Basic Research Program of China (973 Program) (No. 2014CBA02004), the Natural Science Foundation of China (Nos. 91753203, 81773018, 81771138), the Special Project on Precision Medicine under the National Key R&D Program (No. 2017YFC0906600), the Innovation Project of Instrument and Equipment Function Development of the Chinese Academy of Sciences (No. 2060499), Shanghai Rising-Star Program by Science and Technology Commission of Shanghai Municipality (17QA1400800) and K. C. Wong Education Foundation.

## Author contributions

M.T., Y.L., and X.L. conceived the research ideas, supervised the project and wrote the manuscript. B.L., S.J., M.L., and X.X. performed the animal and cellular experiments and analyzed the data. D.L. purified the USP14 complex. M.Z., L.Z., and S.S. performed some of the LC/MS experiments, J.L. and L.Z. provided technical advice on bioinformation. H.L., L.Q., and J.L. provided technical advice and discussed on the cellular studies. M.T. is the guarantor of this work and, as such, had full access to all the data in the study and takes responsibility for the integrity of the data and the accuracy of the data analyses.

## Additional information

**Competing interests:** The authors declare no competing interests.

