## [Peer review file · Nature Communications]

Reviewer #1 (Remarks to the Author):

In the manuscript entitled “Proteome-wide analysis of ubiquitin-specific protease 14 substrates: role as a key modulator of liver steatosis via stabilizing fatty acid synthase in obese mice”, the authors identified a novel substrate of USP14 by proteome, ubiquitinome, and interactome analysis, and further demonstrated that USP14 is involved in hepatic TG metabolism by regulating FASN stability. The technical range of these experiments is laudable, and the findings are of significant interest; However, there are some points that should be addressed.

1. For several in vitro and in vivo experiments, duration of adenovirus-mediated overexpression or knockdown is unclear, as is if the authors sacrificed mice ad libitum, or after fasting or fasting followed by refeeding. This is critical for interpretation of the results, as it will speak directly to whether USP14 increases hepatic TG levels by regulating FASN stability, rather than the insulin/Srebp1c-mediated lipogenesis.

2. The authors show that WT-USP14 increases FASN protein levels, while catalytically inactive (CI) USP14 variant reduces its protein levels, (Figure 6A). Can the authors show that CI-USP14 variant does not affect FASN ubiquitination?

3. Knockdown of USP14 appears to reduce blood glucose and insulin levels, and improve insulin tolerance in db/db mice. This result is not necessarily expected, as liver-specific deletion of ACACA (another key regulator of lipogenesis) shows no effects on glucose homeostasis, and liver-specific FAS k/o mice have a very mild phenotype. These data do not suggest that a simple reduction of liver fat is sufficient to affect liver insulin sensitivity. Does USP14 knockdown in db/db mice improve the sensitivity of hepatic insulin signaling? Is this mediated by FASN as well? Also, does overexpression of USP14 in wild-type mice impair glucose homeostasis or insulin sensitivity?

4. The authors show that LXR regulates the USP14 expression, but should integrate their data with previous findings that LXR can regulate FASN transcription by direct binding on the FASN promoter at the transcriptional level (JBC, Joseph SB et al., 2002; 277(13): 11019-25).

Also, the manuscript was a bit difficult to read. This may be easily fixed, though. The authors found that the mRNA levels of several USP family members were increased in livers from HFD-fed mice based on data from previously conducted Affymetrix arrays (Supplementary Figure 3); these results suggest the relative importance of USP14 among many of its family members in obese livers. These could provide rationale why they focused on USP14, so I'd suggest starting with Figure 3 and Supplementary Figure 3, then Figure 1 and 2 (which could be combined in one figure).

Reviewer #2 (Remarks to the Author):

Liu and coll. performed a comprehensive proteome-wide analysis to identify substrates of the deubiquitinating enzyme USP14, using quantitative proteomics, ubiquitinome and interactome analyses. Among different candidates they focused their study on fatty acid synthase (FAS), a key enzyme involved in de novo hepatic lipogenesis. Using mouse primary hepatocytes and mouse models the authors provided convincing evidence that USP14 and FAS interact and that modifying USP14 expression similarly altered FAS protein levels and hepatic lipid content. FAS is mainly known to be controlled at the transcriptional level and this study reveals an unsuspected role of the USP14 deubiquitinase in the stabilization and action of FAS. The manuscript is well written and easy to follow. However, a number of points need to be addressed to firmly establish that USP14-induced FAS stabilization is an important parameter for a tight control of metabolic homeostasis.

Specific points :

1- Considering the effect of USP14 KD on FAS expression and the role of FAS in the synthesis of membrane lipids, it would be interesting to know whether the HeLa USP-KD cell line exhibit growth defects or morphological alterations when compared to HeLa cells, and whether such alterations could constitute a bias for the proteomic studies.

2- The authors used an adenoviral strategy to specifically modify USP14 expression in mice liver. Although this strategy is very useful to rapidly modify liver gene expression, it suffers from several drawbacks particularly linked to the induced inflammation, especially when high amounts of Ad are used, as in the present work. The authors should indicate how long after the adenoviral infection the animals were studied. They also should add informations on liver inflammation and integrity.

3- Intracellular lipid stores in hepatocytes are regulated by the autophagy process, and inhibition of autophagy is known to increase TG storage in lipid droplets (Singh et al, Nature 2009). As it has

recently been reported that USP14 negatively regulates autophagy (Xu et al, Genes & Dev 2016), it would be interesting to evaluate to what extent this effect could participate to the steatotic phenotype of Ad-USP14 mice liver.

4- FAS is tightly regulated by insulin through the PI3K-Akt signaling pathway. Although this effect occurs mainly at the transcriptional level, it cannot be excluded that post-transcriptional regulation can also happen. Interestingly, the PI3K-Akt signaling pathway was identified by Gene Ontology and KEGG analysis in the up-regulated ubiquitin sites proteomic analysis. This raises the possibility that insulin signaling could be activated in liver from Ad-USP14 mice and participate to liver lipogenesis.

5- In addition to the ITT, which is mainly an indicator of glucose utilization by the skeletal muscle mass, it will be interesting to document glucose tolerance in Ad-shUSP14 mice which will more directly evaluate liver glucose metabolism (endogenous production).

It must be underlined that the relationships between hepatosteatosis and systemic insulin resistance is not as straightforward as claimed by the authors (paragraph Ablation of USP14 in obese mice). Otherwise, they should have observed insulin resistance in Ad-USP14 mice. Do the authors have any index of a modification of insulin sensitivity/resistance in these mice ?

6- Baldini et al (J Mol Biol 2016, 428 : 3295) recently showed that FAS OGlcnAcylation promoted its interaction with the deubiquitinating enzyme USP2A and its stabilization. In the liver, FAS needs to be activated in fed/refed conditions, and it make sense that a nutrient-dependent post-translational modification could interfere with FAS activity. However, this study thus raised the question of the respective expression of USPs in the liver, and of their respective ability to bind to FAS and stabilize the protein.

Other points :

- Figure 4B : the Venn diagram shows that 5 proteins are found by the 3 proteomic approaches, and 4 are listed.
- Suppl Figure 4 : the cellular model used should be indicated in the legend.
- The legend of Figure 5 should be corrected : panel 5B shows only FASN mRNA levels ; panel 5D shows FASN protein and mRNA levels.
- The number of mice studied in Fig.6 and Fig.7 should be indicated.

- Ad-shFAS is lacking in the methods

Reviewer #3 (Remarks to the Author):

The authors have compared the proteome of HeLa cells with knockdown of USP14 to wt cells by SILAC with 217 proteins concluded to be more than 1.2 fold lower in abundance and 76 higher in abundance. Three enzymes in fatty acid synthesis and seven enzymes in fatty acid degradation with lower abundance were concluded to be an indication of a possible role in fatty acid metabolism. With anti diglycine, the tryptic peptides from the SILAC mixture were selected for MS/MS with 449 different tryptic peptides concluded to be increased by >1.2 fold. The authors conclude that 4 enzymes of fatty acid metabolism and the TCA cycle had increased ubiquitin levels and proteins of other pathways are also indicated. The authors reanalysed prior data on mRNA levels of livers of high fat diet mice. They concluded that increased mRNA for USP14 and other USP family members were increased and extended to new data with PCR and western blot. They found USP expression higher in livers from db/db mice and livers from patients with non-alcoholic fatty liver disease. They conclude that mRNA of USP14 correlated with hepatic triglyceride levels.

The authors expressed Flag tagged USP14 in HEK293 cells and conclude that 514 different proteins may be potential interacting proteins.

The authors identified 3 potential substrates of USP14 with two (Fatty acid synthase and creatine kinase B) reduced by western blot of USP14 knockdown cells. The authors show fatty acid synthase in an IP of USP14 in mouse primary hepatocytes. Overexpression of USP14 was concluded to increase fatty acid synthase protein but not mRNA levels while knockdown of USP14 was concluded to decrease fatty acid synthase protein but not mRNA. Expression of USP14 in mouse liver after tail vein injection of adenovirus constructs was concluded to increase liver weight and liver triglyceride levels as well as plasma triglyceride. A combination of an adenovirus construct of shRNA for fatty acid synthase and a construct to express USP14 was concluded to demonstrate after tail vein injection, a block in steatosis. The authors administered IU1 to db/db mice to observe reduced triglyceride and fatty acid synthase levels. The silencing of USP14 expression was attempted with shRNA adenovirus constructs to observe lower fatty acid synthase protein by western blot, lower liver weight, lower triglyceride and to conclude that hepatosteatosis was alleviated along with improved hyperglycemia, hyperinsulinemia, and insulin resistance. The authors conclude that TO901317 treatment of mouse primary hepatocytes increased USP14 expression. Liver X receptor was concluded to increase the transcription of the USP14 gene in mouse primary hepatocytes treated with TO901317.

The authors may consider some suggestions for the individual experiments and coherence. For the SILAC experiments, they may be technical repeats rather than separate cultures. Could the authors do n=3 experiments with different cultures and for at least one, the isotopes are reversed that is put into the cultures for USP 14 knockdowns? As well could the authors do n=3 separate cultures for the experiments with FLAG USP14? For Fig3 are the results from replicates on the same samples with grouped mice or have separate samples from different mice been done n=3 separate times, i.e., genuine biological replicates? For Figs 4,5 can this be done for n=3 biological replicates? Can n=3 be done for Fig6 with averages instead of representative experiments? For Fig 7 can this be extended to 3 groups of mice with statistics and standard deviations instead of SEM? Can Fig 8 be done for n=3 biological replicates?

The coherence of the paper may need to be addressed. The rationale of the cells chosen, the mice selected for study and the strategies employed are not clear to this reviewer nor whether the data actually justify the overall conclusion that USP14 represents a therapeutic target in fatty liver disease and related disorders.

Reviewers' comments:

Reviewer #1 (Remarks to the Author):

In the manuscript entitled “Proteome-wide analysis of ubiquitin-specific protease 14 substrates: role as a key modulator of liver steatosis via stabilizing fatty acid synthase in obese mice”, the authors identified a novel substrate of USP14 by proteome, ubiquitinome, and interactome analysis, and further demonstrated that USP14 is involved in hepatic TG metabolism by regulating FASN stability. The technical range of these experiments is laudable, and the findings are of significant interest; However, there are some points that should be addressed.

1. For several in vitro and in vivo experiments, duration of adenovirus-mediated overexpression or knockdown is unclear, as is if the authors sacrificed mice ad libitum, or after fasting or fasting followed by refeeding. This is critical for interpretation of the results, as it will speak directly to whether USP14 increases hepatic TG levels by regulating FASN stability, rather than the insulin/Srebp1c-mediated lipogenesis.

Reply: We appreciate the comments from Reviewer #1. Duration of adenovirus-mediated overexpression or knockdown has been added in the ‘Figure Legends’ section. For adenovirus-mediated overexpression experiments, C57BL/6 mice were sacrificed at day 10 after a 16-hour fast. For adenovirus-mediated knockdown experiments, *db/db* mice were sacrificed at day 12 after a 16-hour fast.

In addition, we agree with the Reviewer that we should determine whether the steatotic role of USP14 is dependent on insulin. To clarify this issue, C57BL/6 mice were treated with streptozotocin (STZ), which induced insulin deficiency due to the selective pancreatic beta-cell toxicity. 14 days later, mice were administered with adenovirus containing USP14 or GFP via tail vein injection. As a result, overexpression of USP14 increased liver triglyceride contents and FASN protein levels. Therefore, our results indicate that the role of USP14 may be independent on insulin. The results above mentioned have been added in the ‘Supplementary Fig. S6A – S6B’

2. The authors show that WT-USP14 increases FASN protein levels, while catalytically

inactive (CI) USP14 variant reduces its protein levels, (Figure 6A). Can the authors show that CI-USP14 variant does not affect FASN ubiquitination?

Reply: We appreciate the comment. We performed additional *in vitro* experiments as suggested. We showed that the protein levels of FASN gradually decreased in the presence of catalytically inactive USP14, suggesting that the CI-USP14 does not affect FASN stability. The results above mentioned have been added in the 'Figure 5F'

3. Knockdown of USP14 appears to reduce blood glucose and insulin levels, and improve insulin tolerance in db/db mice. This result is not necessarily expected, as liver-specific deletion of ACACA (another key regulator of lipogenesis) shows no effects on glucose homeostasis, and liver-specific FAS k/o mice have a very mild phenotype. These data do not suggest that a simple reduction of liver fat is sufficient to affect liver insulin sensitivity. Does USP14 knockdown in db/db mice improve the sensitivity of hepatic insulin signaling? Is this mediated by FASN as well? Also, does overexpression of USP14 in wild-type mice impair glucose homeostasis or insulin sensitivity?

Reply: We appreciate the comment. Recent genetically engineered mice have demonstrated the contribution of *de novo* fatty acid synthesis to hepatic steatosis and insulin resistance.

1. For the ACC enzyme as indicated by the Reviewer, two liver-specific ACC1-knockout mouse models have been generated (Mao J et al. Proc Natl Acad Sci U S A. 2006; 103:8552-7; Harada N, et al. Mol Cell Biol. 2007; 27:1881-8). However, these models led to moderate phenotypes and did not clearly elucidate the role of ACC1, since, in both cases, a compensatory upregulation in ACC2 expression occurred due to the lack of functional ACC1. An alternative approach to gene knockout models, in which intraperitoneal injection of antisense oligonucleotide (ASO) inhibitors was used to knock down ACC1 and ACC2 expression, either independently or synergistically, helped resolve the distinct roles of these enzymes in the control of *de novo* lipogenesis (Savage DB, et al. J Clin Invest. 2006; 116:817-24). As a result, under high-fat-diet conditions *in vivo*, the synergic inhibition of ACC1 and ACC2 was able to significantly reduce hepatic lipids contents, and improve hepatic insulin sensitivity. ACC1 and ACC2 ASO

treatment also significantly reduced hepatic glucose production during the phase of the euglycemic-hyperinsulinemic clamp study (Savage DB, et al. J Clin Invest. 2006; 116:817-24). This beneficial effect was associated with increases in both Akt and Foxo1 phosphorylation. Altogether, this study has clearly revealed that inhibition of lipogenesis by targeting ACC has beneficial effects on both hepatic steatosis and insulin resistance.

2. For the FASN enzyme, it has been shown that FASN expression levels in the liver are correlated with hepatosteatosis in rodents and humans, suggesting that inhibiting FASN might be a potential target for insulin resistance and liver steatosis (Menendez JA, et al. Clin Chem. 2009; 55:425-38). Indeed, administration of platensimycin, the selective FASN inhibitor, led to a net reduction in liver triglyceride levels and improved insulin sensitivity in *db/+* mice fed a high-fructose diet (Wu M, et al. Proc Natl Acad Sci U S A. 2011; 108:5378-83). Moreover, the benefit efficacy of platensimycin was also generally reproduced in preclinical models with *de novo* lipogenesis tones comparable to humans, including diet-induced obese mice and non-human primates (Singh SB, et al. PLoS One. 2016; 11:e0164133). Therefore, together with our observations, expression or activity of FASN play an important role in the pathogenesis of fatty liver, insulin resistance and related metabolic disorders.
3. We performed additional experiments as suggested by the Reviewer. As shown in the Supplementary Fig. S8A-8B, phosphorylated AKT was enhanced while gluconeogenic genes (PEPCK and G6Pase) were downregulated in Ad-shUSP14-infected *db/db* mice, suggesting that knockdown of USP14 could improve the sensitivity of hepatic insulin signaling in *db/db* mice. Moreover, overexpression of USP14 reduced glucose tolerance and insulin sensitivity in C57BL/6 mice (Supplementary Fig. S5C-5D), while knockdown of endogenous FASN expression largely blocked the roles of USP14 overexpression (Supplementary Fig. S5H-5I), suggesting that the role of USP14 is, at least in part, dependent on up-regulation of FASN.
4. The authors show that LXR regulates the USP14 expression, but should integrate their data with previous findings that LXR can regulate FASN transcription by direct

binding on the FASN promoter at the transcriptional level (JBC, Joseph SB et al., 2002; 277(13): 11019-25).

Reply: We appreciate the comment. We agree with the Reviewer that LXR could directly regulate FASN transcription by binding to its promoter region. Therefore, we conclude that LXR could regulate FASN expression by transcriptional and post-transcriptional level. We have added some discussion about this issue in the 'Discussion' section.

Also, the manuscript was a bit difficult to read. This may be easily fixed, though. The authors found that the mRNA levels of several USP family members were increased in livers from HFD-fed mice based on data from previously conducted Affymetrix arrays (Supplementary Figure 3); these results suggest the relative importance of USP14 among many of its family members in obese livers. These could provide rationale why they focused on USP14, so I'd suggest starting with Figure 3 and Supplementary Figure 3, then Figure 1 and 2 (which could be combined in one figure).

Reply: We appreciate the comment. We have fixed the manuscript as suggested. In our revised manuscript, we start with the aberrant expression of USP14 in obese livers.

Reviewer #2 (Remarks to the Author):

Liu and coll. performed a comprehensive proteome-wide analysis to identify substrates of the deubiquitinating enzyme USP14, using quantitative proteomics, ubiquitinome and interactome analyses. Among different candidates they focused their study on fatty acid synthase (FAS), a key enzyme involved in de novo hepatic lipogenesis. Using mouse primary hepatocytes and mouse models the authors provided convincing evidence that USP14 and FAS interact and that modifying USP14 expression similarly altered FAS protein levels and hepatic lipid content. FAS is mainly

known to be controlled at the transcriptional level and this study reveals an unsuspected role of the USP14 deubiquitinase in the stabilization and action of FAS. The manuscript is well written and easy to follow. However, a number of points need to be addressed to firmly establish that USP14-induced FAS stabilization is an important parameter for a tight control of metabolic homeostasis.

Specific points :

1- Considering the effect of USP14 KD on FAS expression and the role of FAS in the synthesis of membrane lipids, it would be interesting to know whether the HeLa USP-KD cell line exhibit growth defects or morphological alterations when compared to HeLa cells, and whether such alterations could constitute a bias for the proteomic studies.

Reply: We appreciate the comments from Reviewer #2. We have compared the morphology of control and USP14-KD cells by using microscope, and also detected the growth ratio by cell counting assay. We have only observed a very minor change of cell morphological alterations and growth ratios between these two cell lines under normal cell culture condition for more than one week (please see the data below), suggesting that knock down of USP14 is not sufficient to yield a proteomic study bias.

2- The authors used an adenoviral strategy to specifically modify USP14 expression in mice liver. Although this strategy is very useful to rapidly modify liver gene expression, it suffers from several drawbacks particularly linked to the induced inflammation, especially when high amounts of Ad are used, as in the present work. The authors should indicate how long after the adenoviral infection the animals were studied. They also should add informations on liver inflammation and integrity.

Reply: We appreciate the comments. We agree with the Reviewer that adenovirus inject may induce acute liver injury in animals. Duration of adenovirus-mediated overexpression or knockdown has been added in the 'Figure Legends' section. For adenovirus-mediated overexpression experiments, C57BL/6 mice were sacrificed at day 10 after a 16-hour fast. For adenovirus-mediated knockdown experiments, *db/db* mice were sacrificed at day 12 after a 16-hour fast. We also performed additional experiments showing that our adenovirus did not induce liver inflammation and injury, compared with saline. As shown in the Supplementary Fig. S5A-S5B, the Ad-GFP treatment of C57BL/6 mice did not affect plasma ALT, AST levels and

expression of pro-inflammatory cytokines, compared to saline treatment

3- Intracellular lipid stores in hepatocytes are regulated by the autophagy process, and inhibition of autophagy is known to increase TG storage in lipid droplets (Singh et al, Nature 2009). As it has recently been reported that USP14 negatively regulates autophagy (Xu et al, Genes & Dev 2016), it would be interesting to evaluate to what extent this effect could participate to the steatotic phenotype of Ad-USP14 mice liver.

Reply: We appreciate the comments. We performed additional experiments as suggested. Mice were administrated with adenovirus containing USP14 or GFP. 5 days later, mice were treated with 3-methyladenine (3MA), an autophagy inhibitor (Blommaert EF et al. Eur J Biochem. 1997; 243:240-6). As a result, inhibition of autophagy did not reverse the triglyceride accumulation in Ad-USP14 mice (Supplementary Fig. S6C). Therefore, our results suggest that autophagy might not participate the steatotic phenotype of USP14. Although Xu et al. showed that USP14 could negatively regulate autophagy, we speculate that the role of USP14 might be tissue or cell-specific.

4- FAS is tightly regulated by insulin through the PI3K-Akt signaling pathway. Although this effect occurs mainly at the transcriptional level, it cannot be excluded that post-transcriptional regulation can also happen. Interestingly, the PI3K-Akt signaling pathway was identified by Gene Ontology and KEGG analysis in the up-regulated ubiquitin sites proteomic analysis. This raises the possibility that insulin signaling could be activated in liver from Ad-USP14 mice and participate to liver lipogenesis.

Reply: We appreciate the comments. As it was also pointed out by Reviewer #1, we should determine whether the role of USP14 is dependent on insulin. Firstly, in our *in vivo* experiments, mice were fasted overnight and then sacrificed for analysis. Secondly, to clarify this issue, C57BL/6 were treated with streptozotocin (STZ), which induced insulin deficiency due to the selective pancreatic beta-cell toxicity. 14 days later, mice were administrated with adenovirus containing USP14 or GFP via tail vein injection. As a result, overexpression of USP14 increased liver triglyceride contents

and FASN protein levels. Therefore, our results indicate that the role of USP14 may be independent on insulin. The results above mentioned have been added in the 'Supplementary Fig. S6A-6B'

5- In addition to the ITT, which is mainly an indicator of glucose utilization by the skeletal muscle mass, it will be interesting to document glucose tolerance in Ad-shUSP14 mice which will more directly evaluate liver glucose metabolism (endogenous production). It must be underlined that the relationships between hepatosteatosis and systemic insulin resistance is not as straightforward as claimed by the authors (paragraph Ablation of USP14 in obese mice). Otherwise, they should have observed insulin resistance in Ad-USP14 mice. Do the authors have any index of a modification of insulin sensitivity/resistance in these mice?

Reply: We appreciate the comment. We performed additional experiments as suggested by the Reviewer. As shown in the Supplementary Fig. 8A-8B, phosphorylated AKT was enhanced while gluconeogenic genes (PEPCK and G6Pase) were downregulated in Ad-shUSP14-infected *db/db* mice, suggesting that knockdown of USP14 could improve hepatic insulin sensitivity and glucose metabolism in obese mice. Moreover, overexpression of USP14 reduced glucose tolerance and insulin sensitivity in C57BL/6 mice (Supplementary Fig. S5C-5D), while knockdown of endogenous FASN expression largely blocked the roles of USP14 overexpression (Supplementary Fig. S5H-5I), suggesting that the role of USP14 is, at least in part, dependent on up-regulation of FASN.

It has been shown that FASN expression levels in the liver are correlated with hepatosteatosis in rodents and humans, suggesting that inhibiting FASN might be a potential target for insulin resistance and liver steatosis (Menendez JA, et al. Clin Chem. 2009; 55:425-38). Indeed, administration of platensimycin, the selective FASN inhibitor, led to a net reduction in liver triglyceride levels and improved insulin sensitivity in *db/+* mice fed a high-fructose diet (Wu M, et al. Proc Natl Acad Sci U S A. 2011; 108:5378-83). Moreover, the benefit efficacy of platensimycin was also generally reproduced in preclinical models with *de novo* lipogenesis tones comparable to humans, including diet-induced obese mice and non-human primates (Singh SB, et al. PLoS One. 2016; 11:e0164133). Therefore, together with our results,

expression or activity of FASN play an important role in the pathogenesis of fatty liver, insulin resistance and related metabolic disorders.

6- Baldini et al (J Mol Biol 2016, 428 : 3295) recently showed that FAS O-GlcNAcylation promoted its interaction with the deubiquitinating enzyme USP2A and its stabilization. In the liver, FAS needs to be activated in fed/refed conditions, and it make sense that a nutrient-dependent post-translational modification could interfere with FAS activity. However, this study thus raised the question of the respective expression of USPs in the liver, and of their respective ability to bind to FAS and stabilize the protein.

Reply: We appreciate the reviewer's comment on raising the interesting points: Whether other members of USPs could regulate FASN stability? Whether a USP, such as USP14, regulate the FASN activity via the crosstalk between ubiquitination and a nutrient-dependent post-translational modification? These are important questions not only for understanding FASN but also for understanding USPs and protein post-translational modifications. However, addressing these questions requires tremendous efforts. Systematic screening the role of all the 79 members of USPs on FAS activity are pretty time-consuming and costly. Whether there is additional USP that could regulate FASN is an unknown and open question. Furthermore, as a variety of types of PTMs are known to be associated with nutrient and metabolism, such as O-GlcNAcylation, lysine acetylation, S-acylation (Nat Rev Mol Cell Biol. 2017;18:452-465; Nat Rev Mol Cell Biol. 2017; 18:90-101.; Physiol Rev. 2015; 95: 341–376.) and many PTM sites have already been reported on FASN (27 PTM sites including phosphorylation, acetylation, SUMOylation indexed by UniProt database (P49327)). Characterizing the PTMs on FASN, screening and confirming the functional importance of these modifications, and demonstrating their crosstalks and relationships with ubiquitination and UPS14 also requires tremendous efforts, and is an open question as well. The aim of our study is on the system wide screening USP14 substrates with a specific focus on the investigation of its substrate USP14 in liver steatosis. Therefore, we believe, and we hope the reviewer would agree, that these questions are beyond the scope of this study. Nevertheless, following the reviewer's question, we discussed these issues in our revised manuscript:

Yu et al. reported that the Src-homology 2 (SH2) domain-containing tyrosine phosphatase (Shp2) acts as an adaptor molecule connecting ubiquitin E3 ligase COP1 with FASN, thereby regulating FASN ubiquitination and degradation ⁴³. Moreover, USP2a has been shown to interact with and stabilize FASN in prostate cancer cells ⁴⁴. In addition, expression and activity of liver FASN correlate with O-GlcNAcylation contents in obese mice both in a transcription-dependent and -independent manner ⁴⁵. Inhibition of its O-GlcNAc modification increases the interaction between FASN and USP2a in vivo and ex vivo, providing an insight into the control of FASN expression by O-GlcNAcylation ⁴⁵. Thus, both these studies and our findings indicate that characterization of novel regulatory molecules for FASN expression might reveal novel mechanisms into in disease pathogenesis. It also raised the question that whether UPS14, or other liver-enriched deubiquitinase, could also bind to FASN and stabilize the protein via a crosstalk between ubiquitination and a nutrient-dependent post-translational modification, such as O-GlcNAcylation. Further studies are still needed to address these questions.

Other points :

- Figure 4B : the Venn diagram shows that 5 proteins are found by the 3 proteomic approaches, and 4 are listed.

Reply: We apologize for this inconsistency. Following Reviewer #3's suggestion, we carried out more biological replicates on proteome, ubiquitome and interactome experiments. With the new data, our overall results and conclusions are similar. However, we wish to point that more biological replicates from the three independent experiments will increase the confidence of USP14 substrate identification, but will also led to higher false negative results (i.e., more true substrates will be removed). Therefore, our new dataset with more biological replicates led to the identification of 42 potential USP14 substrates (Fig. 4B). FASN and Ubiquitin were identified from all the three datasets. Our detailed response to this question is shown in the Reviewer #3's comments.

- Suppl Figure 4 : the cellular model used should be indicated in the legend.

Reply: We used HeLa cells stably expressing shRNAs against NC or USP14 for the experiment. We added the information in the figure legend.

- The legend of Figure 5 should be corrected : panel 5B shows only FASN mRNA levels ; panel 5D shows FASN protein and mRNA levels.

Reply: We apologize for the mistake and we have revised the Figure Legend.

- The number of mice studied in Fig.6 and Fig.7 should be indicated.

Reply: We have added the information in the Figure legends as suggested.

- Ad-shFAS is lacking in the methods

Reply: We have added the information in the methods as suggested.

Reviewer #3 (Remarks to the Author):

The authors have compared the proteome of HeLa cells with knockdown of USP14 to wt cells by SILAC with 217 proteins concluded to be more than 1.2 fold lower in abundance and 76 higher in abundance. Three enzymes in fatty acid synthesis and seven enzymes in fatty acid degradation with lower abundance were concluded to be an indication of a possible role in fatty acid metabolism. With anti diglycine, the tryptic peptides from the SILAC mixture were selected for MS/MS with 449 different tryptic peptides concluded to be increased by >1.2 fold. The authors conclude that 4 enzymes of fatty acid metabolism and the TCA cycle had increased ubiquitin levels and proteins of other pathways are also indicated. The authors reanalysed prior data on mRNA levels of livers of high fat diet mice. They concluded that increased mRNA for USP14 and other USP family members were increased and extended to new data with PCR and western blot. They found USP expression higher in livers from db/db mice and livers from patients with non-alcoholic fatty liver disease. They conclude that mRNA of USP14 correlated with hepatic triglyceride levels. The authors expressed Flag tagged USP14 in HEK293 cells and conclude that 514 different proteins may be potential interacting proteins. The authors identified 3 potential substrates of USP14 with two (Fatty acid synthase and creatine kinase B) reduced by western blot of USP14 knockdown cells. The authors show fatty acid synthase in an IP of USP14 in mouse primary hepatocytes. Overexpression of USP14 was concluded to increase fatty acid synthase protein but not mRNA levels while knockdown of USP14 was concluded to decrease fatty acid synthase protein but not mRNA. Expression of USP14 in mouse liver after tail vein injection of adenovirus constructs was concluded to increase liver weight and liver triglyceride levels as well as plasma triglyceride. A

combination of an adenovirus construct of shRNA for fatty acid synthase and a construct to express USP14 was concluded to demonstrate after tail vein injection, a block in steatosis. The authors administered IU1 to db/db mice to observe reduced triglyceride and fatty acid synthase levels. The silencing of USP14 expression was attempted with shRNA adenovirus constructs to observe lower fatty acid synthase protein by western blot, lower liver weight, lower triglyceride and to conclude that hepatosteatosis was alleviated along with improved hyperglycemia, hyperinsulinemia, and insulin resistance. The authors conclude that TO901317 treatment of mouse primary hepatocytes increased USP14 expression. Liver X receptor was concluded to increase the transcription of the USP14 gene in mouse primary hepatocytes treated with TO901317.

The authors may consider some suggestions for the individual experiments and coherence. For the SILAC experiments, they may be technical repeats rather than separate cultures. Could the authors do n=3 experiments with different cultures and for at least one, the isotopes are reversed that is put into the cultures for USP 14 knockdowns? As well could the authors do n=3 separate cultures for the experiments with FLAG USP14? For Fig3 are the results from replicates on the same samples with grouped mice or have separate samples from different mice been done n=3 separate times, i.e., genuine biological replicates? For Figs 4,5 can this be done for n=3 biological replicates? Can n=3 be done for Fig6 with averages instead of representative experiments? For Fig 7 can this be extended to 3 groups of mice with statistics and standard deviations instead of SEM? Can Fig 8 be done for n=3 biological replicates?

Reply: We thank the reviewer for comment on the individual experiments and coherence issue. Following the reviewer's suggestion, we carried out additional biological replicate analyses on proteome, ubiquitome and interactome experiments. For the SILAC experiment, we carried out two additional biological replicates (one forward and one reverse labeling) for proteome profiling. Together with our previous two biological replicate data (and with 3 technical replicates), 7,647 protein groups were identified in the four biological replicate analyses. Among them 108 down-regulated proteins (Supplementary Table S1A) and 50 up-regulated proteins

(Supplementary Table S1B) in the USP14 KD group were identified. For ubiquitinome analysis, we carried out an additional biological and technical replicate analysis. Together, we identified 16,086 ubiquitination sites and 406 significant up-regulated ubiquitin sites. The updated results of pathway enrichment analyses of the proteome and ubiquitinome data were largely consistent with our previous results. The updated figures and data were shown in Fig 2, Fig 3, Supplementary Fig. S2 and S3A, Supplementary Table. S1-4 and S6. For interactome analysis, we carried out two additional biological replicates. FASN were successfully identified by all the three biological replicates. We wish to point that, given the relatively low expression of some USP14 substrates, the protein loss during purification process and the limitation of IP/MS method, not every USP14-interacting protein was successfully recovered by all the three replicates. This is a widely-recognized limitation of IP/MS approach. A previous study led by Drs. Wade Harper and Steven Gygi, two pioneers of protein interactome and ubiquitinome studies, reported that biological replicates only showed a 65% overlap with each other and a 35% overlap with EGFP control complexes (Fig 2C, Cell. 2009; 138:389-403). They also argued that reproducibility itself does not accurately specify a candidate interactor (Fig 2E, Cell. 2009; 138:389-403). Indeed, in two recent interactome studies (Cell. 2015; 162:425-440; Nature. 2017; 545:505-509), they carried out technical replicates rather than additional biological replicates. By integrating the result from the additional two biological IP replicates, we defined proteins with a five-fold change of the emPAI score in at least two of the three biological replicates as potential interaction partners of USP14 (Mol Cell Proteomics. 2005 Sep;4(9):1265-72; Mol Cell Proteomics. 2009 Dec;8(12):2770-7.; Nucleic Acids Res. 2017 Dec 8. doi: 10.1093/nar/gkx1226). In such way, we still identified 21 of the 25 previously USP14 reported interacting, suggesting the reproducibility of our methods. As we discussed in our manuscript, we considered a potential substrate of USP14 could be overlapped from two of the three independent datasets (proteome, ubiquitinome and interactome). Our new dataset with more biological replicates led to the identification of 42 potential USP14 substrates (Fig. 4B). Because additional biological replicates were introduced from all the three independent experiments (proteome, ubiquitome and interactome), the number of overlapping proteins and USP14 substrate candidates inevitably

decreased as compared with our previous results. We wish to point that more biological replicates from the three independent experiments will increase the confidence of UPS14 substrate identification, but will also led to higher false negative results (i.e., more true substrates will be removed). For example, both CKB and CEBPB proteins were missed in our updated combined dataset. However, by using independent biochemical experiments (such as immunoprecipitation and Western blot), we clearly found these two proteins could possibly be USP14 substrates (please see the data below for CEBPB and Supplementary Fig. S4 for CKB), although we did not include them in Fig 4B. Nevertheless, in our updated dataset, we can still identify USP14 (our bait protein), ubiquitin itself and FASN from all the experiments, further supporting FASN as a *bona fide* substrate of USP14.

For the animal experiments, representative 2 or 3 lanes of western blots were shown in the Figures 6 and 7. This is a widely accepted way to present the data for animal model studies of metabolic disorders in the literatures (Wang PX, et al. Nat Med. 2017; 23:439-449; Sun S, et al. Nat Cell Biol. 2015; 17:1546-1555; Ma X, et al. Cell Metab. 2015; 22:695-708). In fact, we analyzed more samples and obtained consistent results with Figures 6 and 7. For instance, as shown in the below, protein levels of FASN were reduced in livers of db/db mice injected with two adenoviral shRNAs (shUSP14-1, shUSP14-2), compared with negative control (shNC). This western blot result is consistent with Figure 7A.

The coherence of the paper may need to be addressed. The rationale of the cells chosen, the mice selected for study and the strategies employed are not clear to this reviewer nor whether the data actually justify the overall conclusion that USP14 represents a therapeutic target in fatty liver disease and related disorders.

Reply: We used both 293T and HeLa cells as model cells for proteomic assays. We purified USP14 protein complex in 293T cells which is one of the most widely-used and easily-manipulated cells for such purpose. We used HeLa cells to knockdown USP14, as it contains relatively high level of endogenous USP14 (which could be easily detected by western blot).

For mice model, high-fat-diet-induced obese mice or leptin receptor-deficient mice (db/db) are standard models for obesity, hepatosteatosis, insulin resistance and related metabolic diseases, which are usually employed in literatures (Liu J, et al. Cell. 2016; 167:1052-1066; Awazawa M et al. Nat Med. 2017; 23:1466-1473). Adenovirus-mediated overexpression or knockdown of target genes are also standard strategies for liver research. These models and strategies are used in our previous studies (Wu L, et al. Cell Metab. 2016; 23:735-743; Wang X, et al. J Hepatol. 2015; 63:183-190; Lu Y, et al. J Clin Invest. 2014; 124:3501-3513; Lu Y, et al. Gut. 2014; 63:170-178).

The results we present here are based on several different experimental approaches including cell and mouse studies, and all of the data keep consistent showing that hepatic USP14 plays an important role in the liver steatosis. This was suggested by multiple lines of evidence. First, up-regulation of USP14 is a conserved feature of obesity-related hepatosteatosis. Second, our protein affinity purification identified FASN as a novel interacting protein of USP14. USP14 could enhance FASN stability by

blocking its ubiquitination and degradation. Third, selective overexpression of hepatic USP14 resulted in massive TG accumulation in normal mice, while the hepatic silence of USP14 significantly attenuated hepatosteatosis in db/db mice. Forth, our results demonstrated that FASN could be a direct target of USP14 to exert its function in hepatic lipogenesis. Thus, we propose that USP14 could be an important regulator in obesity-related hepatosteatosis.

Reviewer #4

Under review is the manuscript entitled "Proteome-wide analysis of ubiquitin-specific protease 14 substrates: role as a key modulator of liver steatosis via stabilizing fatty acid synthase in obese mice" by Liu et al. The manuscript describes the application of a multi-pronged proteomic approach to identify novel USP14 substrates, and the evaluation of its role in metabolism. There are several serious concerns that need to be addressed, or may even prevent publication in Nature Communications, a decision by the editor. If the paper is to be reconsidered by Nature Communications, or submitted elsewhere, I would suggest several major modifications to improve the readability, but more importantly, the overall quality and soundness of the claims of the manuscript.

Major comments:

1- The MS database search methodology is presented confusingly and likely inaccurately. The authors should provide all the relevant settings used for processing the MS data. What was the precursor and the fragments mass tolerance? Were any modifications taken into account? What about proteins/peptides FDR? Without this information it is not possible to evaluate the quality of the MS analysis and reproduce the results.

2- In the same light, identifications are done using a 4 year old database (2013). Why not use a more recent one in which genes/proteins are more appropriately annotated. In that way, the identified proteins/sites are more in line with current standards.

3- The authors do not mention the use of any site localization probability filtering, neither do they mention manual interpretation of the MS/MS spectra to verify whether the exact annotation of the site by MaxQuant is accurate. The number of ubiquitination sites is therefore overestimated, please apply proper filtering and change the reported numbers accordingly.

4- Line 161: a 1.2-fold change is generally associated to no biological difference. It is hard to believe such a small change can drive any phenotype. Why using exactly '1.2', when a ~2-fold change is commonly used? Please comment.

5- The shape of the volcano plot (Fig. 1B) looks quite wide, can the authors provide a metric for the quantification reproducibility of the 5 replica?

6- For the proteome and ubiquitome sample preparation, no information is reported about lysis, lysis buffer, and digestion conditions.

7- Lines 361-370: The entire discussion is weak and inaccurate, the authors simply list the well-known technical limitations of the MS-based proteomics workflows. In Supplementary Table S6A 4 proteins are listed as potential USP14 substrates (ABLIM1, AGPAT1, GNAS, IDH2), which have been identified only in the proteome and interactome experiments. Interactors of USP14 are not necessarily also substrates.

8- Supplementary Figure 1, how it is possible that K at position 9 is only at 9%? Did the authors used pre-aligned sequences for the analysis? If so, K at position 9 should have been at ~100%. Is this a zoom-in? Was the entire human proteome used as background? Please clarify

9- It appears to this reviewer that the authors did not provide access to the raw MS data (both MS acquisitions and outputs of Mascot/MaxQuant). I strongly suggest to submit the data to a public repository, such as PRIDE/ProteomeXchange.

Minor comments:

1- Line 562, it is not clear for which data set the 2-way ANOVA was used, please clarify.

2- "... highly possible...", "... dramatically increased..."; please revise the sentences.

3- Line 180: iceLogo, please add the reference.

4- Figure 5B and 5D, Suppl. Fig. 5 and 7, how should the reader interpret the error bars? s.e.m.? s.d.? CI?

Reviewers' comments:

Reviewer #1 (Remarks to the Author):

In the manuscript entitled “Proteome-wide analysis of ubiquitin-specific protease 14 substrates: role as a key modulator of liver steatosis via stabilizing fatty acid synthase in obese mice”, the authors identified a novel substrate of USP14 by proteome, ubiquitinome, and interactome analysis, and further demonstrated that USP14 is involved in hepatic TG metabolism by regulating FASN stability. The technical range of these experiments is laudable, and the findings are of significant interest; However, there are some points that should be addressed.

1. For several in vitro and in vivo experiments, duration of adenovirus-mediated overexpression or knockdown is unclear, as is if the authors sacrificed mice ad libitum, or after fasting or fasting followed by refeeding. This is critical for interpretation of the results, as it will speak directly to whether USP14 increases hepatic TG levels by regulating FASN stability, rather than the insulin/Srebp1c-mediated lipogenesis.

Reply: We appreciate the comments from Reviewer #1. Duration of adenovirus-mediated overexpression or knockdown has been added in the ‘Figure Legends’ section. For adenovirus-mediated overexpression experiments, C57BL/6 mice were sacrificed at day 10 after a 16-hour fast. For adenovirus-mediated knockdown experiments, *db/db* mice were sacrificed at day 12 after a 16-hour fast.

In addition, we agree with the Reviewer that we should determine whether the steatotic role of USP14 is dependent on insulin. To clarify this issue, C57BL/6 mice were treated with streptozotocin (STZ), which induced insulin deficiency due to the selective pancreatic beta-cell toxicity. 14 days later, mice were administered with adenovirus containing USP14 or GFP via tail vein injection. As a result, overexpression of USP14 increased liver triglyceride contents and FASN protein levels. Therefore, our results indicate that the role of USP14 may be independent on insulin. The results above mentioned have been added in the ‘Supplementary Fig. S6A – S6B’

2. The authors show that WT-USP14 increases FASN protein levels, while catalytically inactive (CI) USP14 variant reduces its protein levels, (Figure 6A). Can the authors

show that CI-USP14 variant does not affect FASN ubiquitination?

Reply: We appreciate the comment. We performed additional *in vitro* experiments as suggested. We showed that the protein levels of FASN gradually decreased in the presence of catalytically inactive USP14, suggesting that the CI-USP14 does not affect FASN stability. The results above mentioned have been added in the 'Figure 5F'

3. Knockdown of USP14 appears to reduce blood glucose and insulin levels, and improve insulin tolerance in db/db mice. This result is not necessarily expected, as liver-specific deletion of ACACA (another key regulator of lipogenesis) shows no effects on glucose homeostasis, and liver-specific FAS k/o mice have a very mild phenotype. These data do not suggest that a simple reduction of liver fat is sufficient to affect liver insulin sensitivity. Does USP14 knockdown in db/db mice improve the sensitivity of hepatic insulin signaling? Is this mediated by FASN as well? Also, does overexpression of USP14 in wild-type mice impair glucose homeostasis or insulin sensitivity?

Reply: We appreciate the comment. Recent genetically engineered mice have demonstrated the contribution of *de novo* fatty acid synthesis to hepatic steatosis and insulin resistance.

1. For the ACC enzyme as indicated by the Reviewer, two liver-specific ACC1-knockout mouse models have been generated (Mao J et al. Proc Natl Acad Sci U S A. 2006; 103:8552-7; Harada N, et al. Mol Cell Biol. 2007; 27:1881-8). However, these models led to moderate phenotypes and did not clearly elucidate the role of ACC1, since, in both cases, a compensatory upregulation in ACC2 expression occurred due to the lack of functional ACC1. An alternative approach to gene knockout models, in which intraperitoneal injection of antisense oligonucleotide (ASO) inhibitors was used to knock down ACC1 and ACC2 expression, either independently or synergistically, helped resolve the distinct roles of these enzymes in the control of *de novo* lipogenesis (Savage DB, et al. J Clin Invest. 2006; 116:817-24). As a result, under high-fat-diet conditions *in vivo*, the synergic inhibition of ACC1 and ACC2 was able to significantly reduce hepatic lipids contents, and improve hepatic insulin sensitivity. ACC1 and ACC2 ASO treatment also significantly reduced hepatic glucose production during the phase

of the euglycemic-hyperinsulinemic clamp study (Savage DB, et al. J Clin Invest. 2006; 116:817-24). This beneficial effect was associated with increases in both Akt and Foxo1 phosphorylation. Altogether, this study has clearly revealed that inhibition of lipogenesis by targeting ACC has beneficial effects on both hepatic steatosis and insulin resistance.

2. For the FASN enzyme, it has been shown that FASN expression levels in the liver are correlated with hepatosteatosis in rodents and humans, suggesting that inhibiting FASN might be a potential target for insulin resistance and liver steatosis (Menendez JA, et al. Clin Chem. 2009; 55:425-38). Indeed, administration of platensimycin, the selective FASN inhibitor, led to a net reduction in liver triglyceride levels and improved insulin sensitivity in db/+ mice fed a high-fructose diet (Wu M, et al. Proc Natl Acad Sci U S A. 2011; 108:5378-83). Moreover, the benefit efficacy of platensimycin was also generally reproduced in preclinical models with de novo lipogenesis tones comparable to humans, including diet-induced obese mice and non-human primates (Singh SB, et al. PLoS One. 2016; 11:e0164133). Therefore, together with our observations, expression or activity of FASN play an important role in the pathogenesis of fatty liver, insulin resistance and related metabolic disorders.
3. We performed additional experiments as suggested by the Reviewer. As shown in the Supplementary Fig. S8A-8B, phosphorylated AKT was enhanced while gluconeogenic genes (PEPCK and G6Pase) were downregulated in Ad-shUSP14-infected *db/db* mice, suggesting that knockdown of USP14 could improve the sensitivity of hepatic insulin signaling in *db/db* mice. Moreover, overexpression of USP14 reduced glucose tolerance and insulin sensitivity in C57BL/6 mice (Supplementary Fig. S5C-5D), while knockdown of endogenous FASN expression largely blocked the roles of USP14 overexpression (Supplementary Fig. S5H-5I), suggesting that the role of USP14 is, at least in part, dependent on up-regulation of FASN.
4. The authors show that LXR regulates the USP14 expression, but should integrate their data with previous findings that LXR can regulate FASN transcription by direct binding on the FASN promoter at the transcriptional level (JBC, Joseph SB et al., 2002;

277(13): 11019-25).

Reply: We appreciate the comment. We agree with the Reviewer that LXR could directly regulate FASN transcription by binding to its promoter region. Therefore, we conclude that LXR could regulate FASN expression by transcriptional and post-transcriptional level. We have added some discussion about this issue in the 'Discussion' section.

Also, the manuscript was a bit difficult to read. This may be easily fixed, though. The authors found that the mRNA levels of several USP family members were increased in livers from HFD-fed mice based on data from previously conducted Affymetrix arrays (Supplementary Figure 3); these results suggest the relative importance of USP14 among many of its family members in obese livers. These could provide rationale why they focused on USP14, so I'd suggest starting with Figure 3 and Supplementary Figure 3, then Figure 1 and 2 (which could be combined in one figure).

Reply: We appreciate the comment. We have fixed the manuscript as suggested. In our revised manuscript, we start with the aberrant expression of USP14 in obese livers.

Reviewer #2 (Remarks to the Author):

Liu and coll. performed a comprehensive proteome-wide analysis to identify substrates of the deubiquitinating enzyme USP14, using quantitative proteomics, ubiquitinome and interactome analyses. Among different candidates they focused their study on fatty acid synthase (FAS), a key enzyme involved in de novo hepatic lipogenesis. Using mouse primary hepatocytes and mouse models the authors provided convincing evidence that USP14 and FAS interact and that modifying USP14 expression similarly altered FAS protein levels and hepatic lipid content. FAS is mainly known to be controlled at the transcriptional level and this study reveals an

unsuspected role of the USP14 deubiquitinase in the stabilization and action of FAS. The manuscript is well written and easy to follow. However, a number of points need to be addressed to firmly establish that USP14-induced FAS stabilization is an important parameter for a tight control of metabolic homeostasis.

Specific points :

1- Considering the effect of USP14 KD on FAS expression and the role of FAS in the synthesis of membrane lipids, it would be interesting to know whether the HeLa USP-KD cell line exhibit growth defects or morphological alterations when compared to HeLa cells, and whether such alterations could constitute a bias for the proteomic studies.

Reply: We appreciate the comments from Reviewer #2. We have compared the morphology of control and USP14-KD cells by using microscope, and also detected the growth ratio by cell counting assay. We have only observed a very minor change of cell morphological alterations and growth ratios between these two cell lines under normal cell culture condition for more than one week (please see the data below), suggesting that knock down of USP14 is not sufficient to yield a proteomic study bias.

2- The authors used an adenoviral strategy to specifically modify USP14 expression in mice liver. Although this strategy is very useful to rapidly modify liver gene expression, it suffers from several drawbacks particularly linked to the induced inflammation, especially when high amounts of Ad are used, as in the present work. The authors should indicate how long after the adenoviral infection the animals were studied. They also should add informations on liver inflammation and integrity.

Reply: We appreciate the comments. We agree with the Reviewer that adenovirus inject may induce acute liver injury in animals. Duration of adenovirus-mediated overexpression or knockdown has been added in the 'Figure Legends' section. For adenovirus-mediated overexpression experiments, C57BL/6 mice were sacrificed at day 10 after a 16-hour fast. For adenovirus-mediated knockdown experiments, *db/db* mice were sacrificed at day 12 after a 16-hour fast. We also performed additional experiments showing that our adenovirus did not induce liver inflammation and injury, compared with saline. As shown in the Supplementary Fig. S5A-S5B, the Ad-GFP treatment of C57BL/6 mice did not affect plasma ALT, AST levels and

expression of pro-inflammatory cytokines, compared to saline treatment

3- Intracellular lipid stores in hepatocytes are regulated by the autophagy process, and inhibition of autophagy is known to increase TG storage in lipid droplets (Singh et al, Nature 2009). As it has recently been reported that USP14 negatively regulates autophagy (Xu et al, Genes & Dev 2016), it would be interesting to evaluate to what extent this effect could participate to the steatotic phenotype of Ad-USP14 mice liver.

Reply: We appreciate the comments. We performed additional experiments as suggested. Mice were administrated with adenovirus containing USP14 or GFP. 5 days later, mice were treated with 3-methyladenine (3MA), an autophagy inhibitor (Blommaert EF et al. Eur J Biochem. 1997; 243:240-6). As a result, inhibition of autophagy did not reverse the triglyceride accumulation in Ad-USP14 mice (Supplementary Fig. S6C). Therefore, our results suggest that autophagy might not participate the steatotic phenotype of USP14. Although Xu et al. showed that USP14 could negatively regulate autophagy, we speculate that the role of USP14 might be tissue or cell-specific.

4- FAS is tightly regulated by insulin through the PI3K-Akt signaling pathway. Although this effect occurs mainly at the transcriptional level, it cannot be excluded that post-transcriptional regulation can also happen. Interestingly, the PI3K-Akt signaling pathway was identified by Gene Ontology and KEGG analysis in the up-regulated ubiquitin sites proteomic analysis. This raises the possibility that insulin signaling could be activated in liver from Ad-USP14 mice and participate to liver lipogenesis.

Reply: We appreciate the comments. As it was also pointed out by Reviewer #1, we should determine whether the role of USP14 is dependent on insulin. Firstly, in our *in vivo* experiments, mice were fasted overnight and then sacrificed for analysis. Secondly, to clarify this issue, C57BL/6 were treated with streptozotocin (STZ), which induced insulin deficiency due to the selective pancreatic beta-cell toxicity. 14 days later, mice were administrated with adenovirus containing USP14 or GFP via tail vein injection. As a result, overexpression of USP14 increased liver triglyceride contents

and FASN protein levels. Therefore, our results indicate that the role of USP14 may be independent on insulin. The results above mentioned have been added in the 'Supplementary Fig. S6A-6B'

5- In addition to the ITT, which is mainly an indicator of glucose utilization by the skeletal muscle mass, it will be interesting to document glucose tolerance in Ad-shUSP14 mice which will more directly evaluate liver glucose metabolism (endogenous production). It must be underlined that the relationships between hepatosteatosis and systemic insulin resistance is not as straightforward as claimed by the authors (paragraph Ablation of USP14 in obese mice). Otherwise, they should have observed insulin resistance in Ad-USP14 mice. Do the authors have any index of a modification of insulin sensitivity/resistance in these mice?

Reply: We appreciate the comment. We performed additional experiments as suggested by the Reviewer. As shown in the Supplementary Fig. 8A-8B, phosphorylated AKT was enhanced while gluconeogenic genes (PEPCK and G6Pase) were downregulated in Ad-shUSP14-infected *db/db* mice, suggesting that knockdown of USP14 could improve hepatic insulin sensitivity and glucose metabolism in obese mice. Moreover, overexpression of USP14 reduced glucose tolerance and insulin sensitivity in C57BL/6 mice (Supplementary Fig. S5C-5D), while knockdown of endogenous FASN expression largely blocked the roles of USP14 overexpression (Supplementary Fig. S5H-5I), suggesting that the role of USP14 is, at least in part, dependent on up-regulation of FASN.

It has been shown that FASN expression levels in the liver are correlated with hepatosteatosis in rodents and humans, suggesting that inhibiting FASN might be a potential target for insulin resistance and liver steatosis (Menendez JA, et al. Clin Chem. 2009; 55:425-38). Indeed, administration of platensimycin, the selective FASN inhibitor, led to a net reduction in liver triglyceride levels and improved insulin sensitivity in *db/+* mice fed a high-fructose diet (Wu M, et al. Proc Natl Acad Sci U S A. 2011; 108:5378-83). Moreover, the benefit efficacy of platensimycin was also generally reproduced in preclinical models with *de novo* lipogenesis tones comparable to humans, including diet-induced obese mice and non-human primates (Singh SB, et al. PLoS One. 2016; 11:e0164133). Therefore, together with our results,

expression or activity of FASN play an important role in the pathogenesis of fatty liver, insulin resistance and related metabolic disorders.

6- Baldini et al (J Mol Biol 2016, 428 : 3295) recently showed that FAS O-GlcNAcylation promoted its interaction with the deubiquitinating enzyme USP2A and its stabilization. In the liver, FAS needs to be activated in fed/refed conditions, and it make sense that a nutrient-dependent post-translational modification could interfere with FAS activity. However, this study thus raised the question of the respective expression of USPs in the liver, and of their respective ability to bind to FAS and stabilize the protein.

Reply: We appreciate the reviewer's comment on raising the interesting points: Whether other members of USPs could regulate FASN stability? Whether a USP, such as USP14, regulate the FASN activity via the crosstalk between ubiquitination and a nutrient-dependent post-translational modification? These are important questions not only for understanding FASN but also for understanding USPs and protein post-translational modifications. However, addressing these questions requires tremendous efforts. Systematic screening the role of all the 79 members of USPs on FAS activity are pretty time-consuming and costly. Whether there is additional USP that could regulate FASN is an unknown and open question. Furthermore, as a variety of types of PTMs are known to be associated with nutrient and metabolism, such as O-GlcNAcylation, lysine acetylation, S-acylation (Nat Rev Mol Cell Biol. 2017;18:452-465; Nat Rev Mol Cell Biol. 2017; 18:90-101.; Physiol Rev. 2015; 95: 341–376.) and many PTM sites have already been reported on FASN (27 PTM sites including phosphorylation, acetylation, SUMOylation indexed by UniProt database (P49327)). Characterizing the PTMs on FASN, screening and confirming the functional importance of these modifications, and demonstrating their crosstalks and relationships with ubiquitination and UPS14 also requires tremendous efforts, and is an open question as well. The aim of our study is on the system wide screening USP14 substrates with a specific focus on the investigation of its substrate USP14 in liver steatosis. Therefore, we believe, and we hope the reviewer would agree, that these questions are beyond the scope of this study. Nevertheless, following the reviewer's question, we discussed these issues in our revised manuscript:

Yu et al. reported that the Src-homology 2 (SH2) domain-containing tyrosine phosphatase (Shp2) acts as an adaptor molecule connecting ubiquitin E3 ligase COP1 with FASN, thereby regulating FASN ubiquitination and degradation ⁴³. Moreover, USP2a has been shown to interact with and stabilize FASN in prostate cancer cells ⁴⁴. In addition, expression and activity of liver FASN correlate with O-GlcNAcylation contents in obese mice both in a transcription-dependent and -independent manner ⁴⁵. Inhibition of its O-GlcNAc modification increases the interaction between FASN and USP2a in vivo and ex vivo, providing an insight into the control of FASN expression by O-GlcNAcylation ⁴⁵. Thus, both these studies and our findings indicate that characterization of novel regulatory molecules for FASN expression might reveal novel mechanisms into in disease pathogenesis. It also raised the question that whether UPS14, or other liver-enriched deubiquitinase, could also bind to FASN and stabilize the protein via a crosstalk between ubiquitination and a nutrient-dependent post-translational modification, such as O-GlcNAcylation. Further studies are still needed to address these questions.

Other points :

- Figure 4B : the Venn diagram shows that 5 proteins are found by the 3 proteomic approaches, and 4 are listed.

Reply: We apologize for this inconsistency. Following Reviewer #3's suggestion, we carried out more biological replicates on proteome, ubiquitome and interactome experiments. With the new data, our overall results and conclusions are similar. However, we wish to point that more biological replicates from the three independent experiments will increase the confidence of USP14 substrate identification, but will also led to higher false negative results (i.e., more true substrates will be removed). Therefore, our new dataset with more biological replicates led to the identification of 42 potential USP14 substrates (Fig. 4B). FASN and Ubiquitin were identified from all the three datasets. Our detailed response to this question is shown in the Reviewer #3's comments.

- Suppl Figure 4 : the cellular model used should be indicated in the legend.

Reply: We used HeLa cells stably expressing shRNAs against NC or USP14 for the experiment. We added the information in the figure legend.

- The legend of Figure 5 should be corrected : panel 5B shows only FASN mRNA levels ; panel 5D shows FASN protein and mRNA levels.

Reply: We apologize for the mistake and we have revised the Figure Legend.

- The number of mice studied in Fig.6 and Fig.7 should be indicated.

Reply: We have added the information in the Figure legends as suggested.

- Ad-shFAS is lacking in the methods

Reply: We have added the information in the methods as suggested.

Reviewer #3 (Remarks to the Author):

The authors have compared the proteome of HeLa cells with knockdown of USP14 to wt cells by SILAC with 217 proteins concluded to be more than 1.2 fold lower in abundance and 76 higher in abundance. Three enzymes in fatty acid synthesis and seven enzymes in fatty acid degradation with lower abundance were concluded to be an indication of a possible role in fatty acid metabolism. With anti diglycine, the tryptic peptides from the SILAC mixture were selected for MS/MS with 449 different tryptic peptides concluded to be increased by >1.2 fold. The authors conclude that 4 enzymes of fatty acid metabolism and the TCA cycle had increased ubiquitin levels and proteins of other pathways are also indicated. The authors reanalysed prior data on mRNA levels of livers of high fat diet mice. They concluded that increased mRNA for USP14 and other USP family members were increased and extended to new data with PCR and western blot. They found USP expression higher in livers from db/db mice and livers from patients with non-alcoholic fatty liver disease. They conclude that mRNA of USP14 correlated with hepatic triglyceride levels. The authors expressed Flag tagged USP14 in HEK293 cells and conclude that 514 different proteins may be potential interacting proteins. The authors identified 3 potential substrates of USP14 with two (Fatty acid synthase and creatine kinase B) reduced by western blot of USP14 knockdown cells. The authors show fatty acid synthase in an IP of USP14 in mouse primary hepatocytes. Overexpression of USP14 was concluded to increase fatty acid synthase protein but not mRNA levels while knockdown of USP14 was concluded to decrease fatty acid synthase protein but not mRNA. Expression of USP14 in mouse liver after tail vein injection of adenovirus constructs was concluded to increase liver weight and liver triglyceride levels as well as plasma triglyceride. A

combination of an adenovirus construct of shRNA for fatty acid synthase and a construct to express USP14 was concluded to demonstrate after tail vein injection, a block in steatosis. The authors administered IU1 to db/db mice to observe reduced triglyceride and fatty acid synthase levels. The silencing of USP14 expression was attempted with shRNA adenovirus constructs to observe lower fatty acid synthase protein by western blot, lower liver weight, lower triglyceride and to conclude that hepatosteatosis was alleviated along with improved hyperglycemia, hyperinsulinemia, and insulin resistance. The authors conclude that TO901317 treatment of mouse primary hepatocytes increased USP14 expression. Liver X receptor was concluded to increase the transcription of the USP14 gene in mouse primary hepatocytes treated with TO901317.

The authors may consider some suggestions for the individual experiments and coherence. For the SILAC experiments, they may be technical repeats rather than separate cultures. Could the authors do n=3 experiments with different cultures and for at least one, the isotopes are reversed that is put into the cultures for USP 14 knockdowns? As well could the authors do n=3 separate cultures for the experiments with FLAG USP14? For Fig3 are the results from replicates on the same samples with grouped mice or have separate samples from different mice been done n=3 separate times, i.e., genuine biological replicates? For Figs 4,5 can this be done for n=3 biological replicates? Can n=3 be done for Fig6 with averages instead of representative experiments? For Fig 7 can this be extended to 3 groups of mice with statistics and standard deviations instead of SEM? Can Fig 8 be done for n=3 biological replicates?

Reply: We thank the reviewer for comment on the individual experiments and coherence issue. Following the reviewer's suggestion, we carried out additional biological replicate analyses on proteome, ubiquitome and interactome experiments. For the SILAC experiment, we carried out two additional biological replicates (one forward and one reverse labeling) for proteome profiling. Together with our previous two biological replicate data (and with 3 technical replicates), 7,647 protein groups were identified in the four biological replicate analyses. Among them 108 down-regulated proteins (Supplementary Table S1A) and 50 up-regulated proteins

(Supplementary Table S1B) in the USP14 KD group were identified. For ubiquitinome analysis, we carried out an additional biological and technical replicate analysis. Together, we identified 15,241 ubiquitination sites and 392 significant up-regulated ubiquitin sites. The updated results of pathway enrichment analyses of the proteome and ubiquitinome data were largely consistent with our previous results. The updated figures and data were showed in Fig 2, Fig 3, Supplementary Fig. S2 and S3A, Supplementary Table. S1-4 and S6. For interactome analysis, we carried out two additional biological replicates. FASN were successfully identified by all the three biological replicates. We wish to point that, given the relatively low expression of some USP14 substrates, the protein loss during purification process and the limitation of IP/MS method, not every USP14-interacting protein was successfully recovered by all the three replicates. This is a widely-recognized limitation of IP/MS approach. A previous study led by Drs. Wade Harper and Steven Gygi, two pioneers of protein interactome and ubiquitinome studies, reported that biological replicates only showed a 65% overlap with each other and a 35% overlap with EGFP control complexes (Fig 2C, Cell. 2009; 138:389-403). They also argued that reproducibility itself does not accurately specify a candidate interactor (Fig 2E, Cell. 2009; 138:389-403). Indeed, in two recent interactome studies (Cell. 2015; 162:425-440; Nature. 2017; 545:505-509), they carried out technical replicates rather than additional biological replicates. By integrating the result from the additional two biological IP replicates, we defined proteins with a five-fold change of the emPAI score in at least two of the three biological replicates as potential interaction partners of USP14 (Mol Cell Proteomics. 2005 Sep;4(9):1265-72; Mol Cell Proteomics. 2009 Dec;8(12):2770-7.; Nucleic Acids Res. 2017 Dec 8. doi: 10.1093/nar/gkx1226). In such way, we still identified 21 of the 25 previously USP14 reported interacting, suggesting the reproducibility of our methods. As we discussed in our manuscript, we considered a potential substrate of USP14 could be overlapped from two of the three independent datasets (proteome, ubiquitinome and interactome). Our new dataset with more biological replicates led to the identification of 42 potential USP14 substrates (Fig. 4B). Because additional biological replicates were introduced from all the three independent experiments (proteome, ubiquitome and interactome), the number of overlapping proteins and USP14 substrate candidates inevitably

decreased as compared with our previous results. We wish to point that more biological replicates from the three independent experiments will increase the confidence of USP14 substrate identification, but will also led to higher false negative results (i.e., more true substrates will be removed). For example, both CKB and CEBPB proteins were missed in our updated combined dataset. However, by using independent biochemical experiments (such as immunoprecipitation and Western blot), we clearly found these two proteins could possibly be USP14 substrates (please see the data below for CEBPB and Supplementary Fig. S4 for CKB), although we did not include them in Fig 4B. Nevertheless, in our updated dataset, we can still identify USP14 (our bait protein), ubiquitin itself and FASN from all the experiments, further supporting FASN as a *bona fide* substrate of USP14.

For the animal experiments, representative 2 or 3 lanes of western blots were shown in the Figures 6 and 7. This is a widely accepted way to present the data for animal model studies of metabolic disorders in the literatures (Wang PX, et al. Nat Med. 2017; 23:439-449; Sun S, et al. Nat Cell Biol. 2015; 17:1546-1555; Ma X, et al. Cell Metab. 2015; 22:695-708). In fact, we analyzed more samples and obtained consistent results with Figures 6 and 7. For instance, as shown in the below, protein levels of FASN were reduced in livers of db/db mice injected with two adenoviral shRNAs (shUSP14-1, shUSP14-2), compared with negative control (shNC). This western blot result is consistent with Figure 7A.

The coherence of the paper may need to be addressed. The rationale of the cells chosen, the mice selected for study and the strategies employed are not clear to this reviewer nor whether the data actually justify the overall conclusion that USP14 represents a therapeutic target in fatty liver disease and related disorders.

Reply: We used both 293T and HeLa cells as model cells for proteomic assays. We purified USP14 protein complex in 293T cells which is one of the most widely-used and easily-manipulated cells for such purpose. We used HeLa cells to knockdown USP14, as it contains relatively high level of endogenous USP14 (which could be easily detected by western blot).

For mice model, high-fat-diet-induced obese mice or leptin receptor-deficient mice (db/db) are standard models for obesity, hepatosteatosis, insulin resistance and related metabolic diseases, which are usually employed in literatures (Liu J, et al. Cell. 2016; 167:1052-1066; Awazawa M et al. Nat Med. 2017; 23:1466-1473). Adenovirus-mediated overexpression or knockdown of target genes are also standard strategies for liver research. These models and strategies are used in our previous studies (Wu L, et al. Cell Metab. 2016; 23:735-743; Wang X, et al. J Hepatol. 2015; 63:183-190; Lu Y, et al. J Clin Invest. 2014; 124:3501-3513; Lu Y, et al. Gut. 2014; 63:170-178).

The results we present here are based on several different experimental approaches including cell and mouse studies, and all of the data keep consistent showing that hepatic USP14 plays an important role in the liver steatosis. This was suggested by multiple lines of evidence. First, up-regulation of USP14 is a conserved feature of obesity-related hepatosteatosis. Second, our protein affinity purification identified FASN as a novel interacting protein of USP14. USP14 could enhance FASN stability by

blocking its ubiquitination and degradation. Third, selective overexpression of hepatic USP14 resulted in massive TG accumulation in normal mice, while the hepatic silence of USP14 significantly attenuated hepatosteatosis in db/db mice. Forth, our results demonstrated that FASN could be a direct target of USP14 to exert its function in hepatic lipogenesis. Thus, we propose that USP14 could be an important regulator in obesity-related hepatosteatosis.

Reviewer #4 (Remarks to the Author):

Under review is the manuscript entitled “Proteome-wide analysis of ubiquitin-specific protease 14 substrates: role as a key modulator of liver steatosis via stabilizing fatty acid synthase in obese mice” by Liu et al. The manuscript describes the application of a multi-pronged proteomic approach to identify novel USP14 substrates, and the evaluation of its role in metabolism.

There are several serious concerns that need to be addressed, or may even prevent publication in Nature Communications, a decision by the editor. If the paper is to be reconsidered by Nature Communications, or submitted elsewhere, I would suggest several major modifications to improve the readability, but more importantly, the overall quality and soundness of the claims of the manuscript.

Reply: We thank the reviewer’s comments. Following the reviewer’s suggestion, we believe that the proteomics experiment in our revised manuscript has been improved and more clearly presented.

Major comments:

1- The MS database search methodology is presented confusingly and likely inaccurately. The authors should provide all the relevant settings used for processing the MS data. What was the precursor and the fragments mass tolerance? Were any modifications taken into account? What about proteins/peptides FDR? Without this information it is not possible to evaluate the quality of the MS analysis and reproduce the results.

Reply: We thank the reviewer’s comments, and our mass spectrometry data processing was mainly used the MaxQuant’s default parameters. We added the

following information into the revised manuscript: The precursor mass tolerance was 10 ppm for Mascot software and 20 ppm for Maxquant software; and the fragments mass tolerance was 0.5 Da. The modification searching parameters were set as follows: Carbamidomethyl(C) were set as fixed modification, and Acetyl (Protein N-term), GlyGly (K), Oxidation (M), Label: 13C6-Lys and Label: 13C615N4-Arg as variable modifications. The proteins and peptides FDR were 1% for MaxQuant software.

2- In the same light, identifications are done using a 4 year old database (2013). Why not use a more recent one in which genes/proteins are more appropriately annotated. In that way, the identified proteins/sites are more in line with current standards.

Reply: We thank the reviewer's comments. We used UniProt human sequence database of 2013 when we started this project. According to the reviewer's suggestion, we also downloaded the most recent version of human sequence database of 2017. The size and sequence of the two versions of the database are similar (as shown in the following table). We carried out database searching using the new protein sequence database by MaxQuant using one of our proteomics dataset. The results showed that identified proteins and peptides using the two versions of human sequence databases were almost identical (shown in the following table). Similar to our study, a relatively old human sequence database is widely used for mass spectrometry analysis in quite a lot recent studies such as (Bar-Peled L et al. Cell. 2017 Oct 19;171(3):696-709; Kusebauch U et al. Cell. 2016 Jul 28;166(3):766-778.). As there is no obvious difference between the results using the two different version of databases (2013 and 2017), we therefore used the 2013 version for all the mass spectrometry data analysis.

The results using the two database version by MaxQuant (bio-replicate 3)

Database Version	2013	2017
Identified proteins	6,548	6,564
Quantified proteins	5,743	5,747
Up-regulated proteins	518	517
Down-regulated proteins	641	643

3- The authors do not mention the use of any site localization probability filtering, neither do they mention manual interpretation of the MS/MS spectra to verify whether the exact annotation of the site by MaxQuant is accurate. The number of ubiquitination sites is therefore overestimated, please apply proper filtering and change the reported numbers accordingly.

Reply: We appreciate the reviewer's comments on this issue. We are sorry about that we did not filter the localization probability for ubiquitin site identification. After filtering with the localization probability, we identified 15,241 (16,086 in our previous manuscript) ubiquitination sites, 6,562 (6,833 in our previous manuscript) ubiquitination sites of which were quantifiable. We finally obtained 392 (406 in our previous manuscript) significant up-regulated ubiquitination sites (≥ 1.2 -fold change and $p < 0.05$ by student t-test). There is a slight decrease (about 5%) of the identified sites. The results were updated in our revised manuscript. The overall results and conclusion were similar.

4- Line 161: a 1.2-fold change is generally associated to no biological difference. It is hard to believe such a small change can drive any phenotype. Why using exactly '1.2', when a ~ 2 -fold change is commonly used? Please comment.

Reply: We thank the reviewer's comments. First, to identify the differential proteins impacted by USP14, we considered the statistically different proteins ($p < 0.05$) between WT and USP14 KD cells, which is commonly used for many proteomics studies (Bigaud E et al. Mol Cell Proteomics. 2016 May;15(5):1498-510; Aretz I et al. Mol Cell Proteomics. 2016 May;15(5):1526-38.). Second, to further remove the proteins with minor biological difference, we additionally used 1.2 fold of change as a cutoff, which is the SD of our proteomics data as shown in the following figure (Drabovich AP et al. Mol Cell Proteomics. 2016 Jun;15(6):2093-107; Kamkina P et al. Mol Cell Proteomics. 2016 May;15(5):1670-80.). Third, as we have also discussed it in our response to reviewer #3 and in our discussion, we considered a potential substrate of USP14 would be overlapped from two of the three independent datasets (proteome, ubiquitinome and interactome) rather than a single dataset alone. Such a criterion was likely to further filter out the substrates with negligible biological effects. Actually, as additional biological replicates were introduced from all the three

independent experiments, the number of overlapping proteins and USP14 substrate candidates inevitably decreased in our new dataset (Fig. 4B). Such criteria with new biological replicates from the three independent experiments actually increased higher false negative results (i.e., more true substrates will be removed). For example, our independent biochemical experiments showed that CKB and CEBPB could be potential substrates of USP14 (Please see our detailed response to reviewer #3), which were missed in our updated combined dataset. Therefore, we used these criteria together (replicates, p value, 1.2 fold of change, and the overlap between the three independent datasets) for USP14 substrate identification.

5- The shape of the volcano plot (Fig. 1B) looks quite wide, can the authors provide a metric for the quantification reproducibility of the 5 replica?

Reply: We thank the reviewer's comments. According to reviewer # 3's comments, we carried out additional biological replicate experiments (Forward and reverse labelling). With the new data, the shape of the volcano plot was also improved. We found that correlation between the technical replicates were better than biological replicates as shown in the figure below. We wish to point out that as the cellular proteome turnover of the biological replicate would unavoidably be some different if samples were prepared in different time (or environment), especially for such purpose, like our proteome and ubiquitinome data. Therefore, we believe that it would be biologically more meaningful to see the trend of protein change in our study, as our primary focus is on the identification of degradation substrates impacted by UPS14 rather than its exact degradation ratio.

Spearman correlation coefficient

(Exp1 has three technical replicates and Exp2 has two technical replicates)

6- For the proteome and ubiquitome sample preparation, no information is reported about lysis, lysis buffer, and digestion conditions.

Reply: We added the following information in our manuscript: All samples were washed three times with ice-cold Dulbecco's PBS (Mediatech Inc., Manassas, VA), lysed in chilled lysis buffer (8.0 m urea in 0.1 m NH₄HCO₃ supplemented with 1× protease inhibitor cocktail (Calbiochem, Darmstadt, Germany), and incubated on ice for half an hour. After sonication, the debris was removed by centrifugation and the supernatant was collected. The protein concentration of the supernatant was determined by BCA assay. Extracted protein (200 µg) of each sample was reduced with 5 mm DTT (Acros, Belgium) at 56 °C for 30 min. Then, the samples were incubated with 15 mm iodoacetamide (IAA) (Acros, Belgium) at room temperature in the darkness for 30 min and quenched by 15 mm DTT. Samples were digested by trypsin (an enzyme-to-substrate ratio of 1:50) at 37 °C overnight and then digested by trypsin (an enzyme-to-substrate ratio of 1:100) for additional 4 h.

7- Lines 361-370: The entire discussion is weak and inaccurate, the authors simply list the well-known technical limitations of the MS-based proteomics workflows. In Supplementary Table S6A 4 proteins are listed as potential USP14 substrates (ABLIM1,

AGPAT1, GNAS, IDH2), which have been identified only in the proteome and interactome experiments. Interactors of USP14 are not necessarily also substrates.

Reply: We appreciate the reviewer's comments on the technical parts of USP14 substrate identification. Although the limitations of these technologies could be familiar to proteomics researchers, we think these limitations are likely to be unfamiliar to scientists in the field of metabolic disorders or scientists in other fields. Therefore, the authors of this manuscript responsible for NASFLD studies suggested such discussion in our manuscript. We believe that it would be helpful to readers who is not familiar with proteomics technology details. We agree with the reviewer that interactors of USP14 are not necessarily also substrates. Nevertheless, immunoprecipitation (IP) approach is probably the most-widely used biochemical approach and a critical step toward the characterization of the substrates of an enzyme (Nature. 2012 Jan 5;481(7379):90-3; Nature. 2017 Jun 22;546(7659):554-558). For example, a previous study led by Drs. Wade Harper and Steven Gygi (Fig 2C, Cell. 2009; 138:389-403) reported the deubiquitinating enzyme interaction landscape using IP approach. Many of the identified interactors were known to be the substrates of USP (such as ADRM1, TXNL1, PSMA6, PSMB5, etc.). In addition to its direct substrate, the interacting partners could also possibly include upstream regulating proteins, such as kinases and phosphatases and indirect protein complex components, such as proteasome associated proteins. Therefore, to rule out these possibilities, we carried out additional proteome and ubiquitinome experiments, and considered the overlap of the two of the three independent datasets as a potential substrate.

8- Supplementary Figure 1, how it is possible that K at position 9 is only at 9%? Did the authors used pre-aligned sequences for the analysis? If so, K at position 9 should have been at ~100%. Is this a zoom-in? Was the entire human proteome used as background? Please clarify

Reply: We apologize for the misleading of the figure. We showed a zoom-in figure, and K at position 9 is actually at ~100%. We have changed the figure in the revised manuscript now. The up-regulated ubiquitin sequences were used for the analysis, and the entire human proteome was used as background. We added the information in the method section.

9- It appears to this reviewer that the authors did not provide access to the raw MS data (both MS acquisitions and outputs of Mascot/MaxQuant). I strongly suggest to submit the data to a public repository, such as PRIDE/ProteomeXchange.

Reply: We appreciate the reviewer's comments. We added data the data deposition information in our manuscript as follows: All mass spectrometry raw data have been deposited to the iProX Consortium with the dataset identifier "USP14" (Subproject ID: IPX0001138000, <http://iprox.org/page/SSV024.html?url=1516238864132fn9S>, Password: OxRI).

Minor comments:

1- Line 562, it is not clear for which data set the 2-way ANOVA was used, please clarify.

Reply: 2-way ANOVA was used for data analysis in the Figure 6G and 6H, which contain four groups of mice. We have added this information in the figure legend.

2- "... highly possible...", "... dramatically increased..."; please revise the sentences.

Reply: We revised these sentences according to the reviewer's suggestion.

3- Line 180: iceLogo, please add the reference.

Reply: We added the reference (Nature Methods 2009, 6, 786-787).

4- Figure 5B and 5D, Suppl. Fig. 5 and 7, how should the reader interpret the error bars? s.e.m.? s.d.? CI?

Reply: The error bars are SEM. We added the information in the figure legend.

Reviewer #1 (Remarks to the Author):

The authors have responded well to my comments, and the manuscript is substantially improved. I have only a few more minor suggested edits, for clarity:

1. The author shows that phosphorylated Akt was increased in Ad-shUSP14-infected db/db mice. Which site of Akt (T308 or S473) did the author measure? In this figure, total Akt is mislabeled as t-Akp.
2. The autophagy experiment - first, the inhibitor which the author used is 3-MA, not 3-MC in the page 12. Supplementary Fig S6C is mislabeled. What is the difference between the gray and black bars?
3. On page 13, Ad-shUSP14 was mislabeled as ad-shSUP14.
4. The author should provide the method for isolating primary hepatocytes.

Reviewer #2 (Remarks to the Author):

Thanks to the authors for additional data that improved their manuscript.

However, some points remain to be addressed, as detailed below.

Thanks to the authors for adding the markers of toxicity and inflammation in Ad-GFP injected mice. However, it should be important to indicate also these markers for the experimental groups Ad USP14 and Ad shUSP14.

A greater care should be taken to be more accurate in a number of experimental procedures (the following list may not be not exhaustive):

- the protocol for mouse primary hepatocytes preparation is lacking
- clinical and biological parameters of NAFLD and control patients should be indicated

- the protocols used for glucose and insulin tolerance tests should be indicated
- the amount of USP14 shRNA adenovirus used also should be indicated
- in legends of Fig 5, Fig S4, Fig S6, ... the protocol of the WB is unclear as it is described more as an IP experiment (« the lysates were incubated with antibodies and the immunocomplexes were detected by WB with the indicated endogenous antibodies »)
- STZ protocol used : amount of STZ, which controls are used, characterization of the mice (blood glucose, BW modifications, ...). How long after the Ad-USP14 injection the mice were studied.
- The protocol for the 3-MA treatment in vivo in mice is lacking, and the reference indicated in the answer to the reviewers is incorrect (Blommaert et al 1997 refers to in vitro experiments on rat hepatocytes).

In Suppl Fig S4 CKB does not seem to be significantly decreased in HeLa cells expressing shRNA against USP14 (the control β -actin is also slightly decreased). Supporting this, CKB was not indicated in Fig 4B showing the potential substrates of USP14. However, it must be noticed that CKB was previously identified with FASN and UBB as a possible USP14 substrate in the first version of the manuscript. This difference deserves clarification.

Line 300 : change « systematic insulin resistance » into « systemic insulin resistance ».

Line 301 : add « glucose intolerance » (to refer to the glucose tolerance test).

Line 320 : the reference of USP14 in autophagy should be indicated (Xu et al 2016).

Lines 442 and 445: reference 45 seems inappropriately cited.

In Suppl Fig S6 panel C there is an error in the 3-MA legend.

In Suppl Fig S8 add in the legend that the analyses in panels A and B were performed on liver samples.

Line 344 : add also liver after mice.

Reviewer #3 (Remarks to the Author):

The authors have increased the number of biological replicates for the mass spec experiments. It had been hoped this would extend to the remaining experiments although not explicitly stated in the critique. There remains a lot to go through and with so many experiments with a plethora of different model systems the question is how rigorous should the data be?

For example, in supplementary table 1, the data from the replicates are shown with several indicating NA even for 2 of the 4 replicates (e.g. VLDLR) yet this has not impacted the average or the P value. The authors may consider that they summarize all data points in their figures and perhaps change from histograms and SEM to something more rigorous and accurate. This may alleviate any concerns that the data have been selected rather than simply reported with all data used for averages, P values etc. The same criticism may apply to supplementary Table 2 and Table S5 for FASN. Fig 4B indicates that FASN is identified in all 3 data sets but not in all experiments for one of the data sets. Could the authors also consider a more detailed legend for the supplementary tables especially indicating exactly how the replicates are averaged and how all the NA values are considered? Perhaps a definition of NA would be helpful as well?

The authors now may also wish to consider if all the other experiments in the paper should have n=3 biological replicates with possibly quantification of western blots and assurance that the signal is linear etc. One example is Figs7C that may need more experiments and quantification.

The authors are well aware of the current trend away from histograms and SEM to box plots and this may also be considered.

Reviewer #4 (Remarks to the Author):

Under review is the revised manuscript by Liu et al., entitled "Proteome-wide analysis of ubiquitin-specific protease 14 substrates: role as a key modulator of liver steatosis via stabilizing fatty acid synthase in obese mice".

Despite the authors have responded to all the comments, there are still few concerns that need to be addressed:

1- The authors claim that the fragments mass tolerance used in the MS database search was set to 0.5 Da. After inspection of the mqpar.xml file submitted by the authors to the repository iProX, it looks like they have reported the wrong value. As the MS2 spectra have been acquired in high-resolution, the fragments mass tolerance used by MaxQuant was 20 ppm. Please, change the text accordingly.

2- On the same note, as no Mascot .dat files have been submitted to the repository, what was the value used in the Mascot search? 0.5 Da? If so, such a relaxing fragments m/z error window raises the chance for random match and weakens protein identification accuracy. Please, clarify and submit to iProx also the mascot output .dat files. A more appropriate mass tolerance for Orbitrap MS/MS spectra is in the range of 10-20 ppm or 0.05-0.02 Da.

3- The iProX submission looks incomplete. The mqpar.xml for the ubiquitome analysis is missing. Please, submit also the unprocessed .txt files output of MaxQuant. Again, without such information/file it is not possible to fully evaluate the quality of the MS (data)analysis

4- 8.0 m urea should be 8.0 M urea, 15 mm iodoacetamide should be 15 mM iodoacetamide... please use the right symbol to indicate molar concentration (M) throughout the manuscript.

Reviewers' comments:

Reviewer #1 (Remarks to the Author):

The authors have responded well to my comments, and the manuscript is substantially improved. I have only a few more minor suggested edits, for clarity:

1. The author shows that phosphorylated Akt was increased in Ad-shUSP14-infected db/db mice. Which site of Akt (T308 or S473) did the author measure? In this figure, total Akt is mislabeled as t-Akp.

Reply: We appreciate the comment. Serine 473 site of AKT was analyzed in our data. We have added this information and corrected the mislabeled AKT in the revised manuscript.

2. The autophagy experiment - first, the inhibitor which the author used is 3-MA, not 3-MC in the page 12. Supplementary Fig S6C is mislabeled. What is the difference between the gray and black bars?

Reply: We appreciate the comment and apologize for our mistake. We have corrected the word and the label of Supplementary Fig 6C.

3. On page 13, Ad-shUSP14 was mislabeled as ad-shSUP14.

Reply: We appreciate the comment and apologize for our mistake. We have corrected the word.

4. The author should provide the method for isolating primary hepatocytes.

Reply: We appreciate the comment and have added the method in the 'Methods' section as suggested.

Reviewer #2 (Remarks to the Author):

Thanks to the authors for additional data that improved their manuscript.

However, some points remain to be addressed, as detailed below.

Thanks to the authors for adding the markers of toxicity and inflammation in Ad-GFP injected mice. However, it should be important to indicate also these markers for the experimental groups Ad USP14 and Ad shUSP14.

Reply: We appreciate the comment. We performed additional experiments showing that our adenovirus (Ad-GFP, Ad-USP14, Ad-shUSP14) did not induce liver inflammation and injury, compared with saline. Please see the results in the Supplementary Figure 5A-5B, Supplementary Figure 8A-8B.

A greater care should be taken to be more accurate in a number of experimental procedures (the following list may not be not exhaustive):

- the protocol for mouse primary hepatocytes preparation is lacking

Reply: We appreciate the comment and have added the method in the 'Methods' section as suggested.

- clinical and biological parameters of NAFLD and control patients should be indicated

Reply: We appreciate the comment and have added the information in the 'Methods' section as suggested.

- the protocols used for glucose and insulin tolerance tests should be indicated

Reply: We appreciate the comment and have added the information in the 'Methods' section as suggested.

- the amount of USP14 shRNA adenovirus used also should be indicated

Reply: We appreciate the comment and have added the information in the 'Methods' section as suggested.

- in legends of Fig 5, Fig S4, Fig S6, ... the protocol of the WB is unclear as it is described more as an IP experiment (« the lysates were incubated with antibodies and the immunocomplexes were detected by WB with the indicated endogenous antibodies »)

Reply: We appreciate the comment and have revised the Figure Legends as suggested.

- STZ protocol used: amount of STZ, which controls are used, characterization of the mice (blood glucose, BW modifications, ...). How long after the Ad-USP14 injection the mice were studied.

Reply: We appreciate the comment and have added the information in the 'Methods' section as suggested.

- The protocol for the 3-MA treatment in vivo in mice is lacking, and the reference indicated in the answer to the reviewers is incorrect (Blommaert et al 1997 refers to in vitro experiments on rat hepatocytes).

Reply: We appreciate the comment and apologize for mistakenly cited

the reference. We have added the protocol for the 3-MA treatment in the 'Methods' section as suggested.

In Suppl Fig S4 CKB does not seem to be significantly decreased in HeLa cells expressing shRNA against USP14 (the control β -actin is also slightly decreased). Supporting this, CKB was not indicated in Fig 4B showing the potential substrates of USP14. However, it must be noticed that CKB was previously identified with FASN and UBB as a possible USP14 substrate in the first version of the manuscript. This difference deserves clarification.

Reply: We appreciate the comment, and clarified it in our manuscript as follows: The result showed that protein contents of FASN were significantly reduced in the USP14 KD cells compared with control cells, whereas CKB (identified only in one IP replicate) showed mild decrease (Supplementary Figure 4). Therefore, FASN was considered as a possible USP14 candidate, whereas CKB was not included in our potential USP14 substrates in Fig 4B according to our criteria.

Line 300 : change « systematic insulin resistance » into « systemic insulin resistance ».

Reply: We appreciate the comment and have revised the word.

Line 301 : add « glucose intolerance » (to refer to the glucose tolerance test).

Reply: We appreciate the comment and have added the word.

Line 320 : the reference of USP14 in autophagy should be indicated (Xu et al 2016).

Reply: We appreciate the comment and have added the reference.

Lines 442 and 445: reference 45 seems inappropriately cited.

Reply: We appreciate the comment and have revised the reference.

In Suppl Fig S6 panel C there is an error in the 3-MA legend.

Reply: We appreciate the comment and have revised the label.

In Suppl Fig S8 add in the legend that the analyses in panels A and B were performed on liver samples.

Reply: We appreciate the comment and have revised the legend of Supplementary Figure 8A-8B.

Line 344: add also liver after mice.

Reply: We appreciate the comment and have added the word.

Reviewer #3 (Remarks to the Author):

The authors have increased the number of biological replicates for the mass spec experiments. It had been hoped this would extend to the remaining experiments although not explicitly stated in the critique. There remains a lot to go through and with so many experiments with a plethora of different model systems the question is how rigorous should the data be?

Reply: We appreciate the Reviewer's comments. Experiments in several different fields were employed in our study, including proteomics, molecular biology and animal model studies for metabolic disorders.

Therefore, multiple different model systems among these fields were used, as commented by the Reviewer. We believe that these model systems and the way to present the data are commonly used in these studies. In addition to biological replication, our main findings were supported by 2 or more independent experiments. For example, we considered the overlap from two independent experiments of proteome, ubiquitome, and interactome as a potential USP14 candidates. To validate FASN as a direct USP14 substrate, we carried out USP14 knockdown, overexpression, and prepared the catalytically inactive (CI) mutant for biochemistry and molecular biology experiment, in addition to the biological replication of each experiment. To investigate role of USP14 in hepatosteatosis mouse model, we examined and validated the biological consequence of USP14 from adenovirus-mediated USP14 overexpression, USP14 ablation, and the pharmacological inhibition of USP14 by the small molecule - IU1. Therefore, the results we present here are based on several different experimental approaches and all of the data keep consistent showing that hepatic USP14 could promote hepatosteatosis through regulation of FASN stability. We think that cross-validation among these independent experiments, together with replication experiments, can well support our results.

For example, in supplementary table 1, the data from the replicates are shown with several indicating NA even for 2 of the 4 replicates (e.g.VLDLR) yet this has not impacted the average or the P value. The authors may consider that they summarize all data points in their figures and perhaps change from histograms and SEM to something more rigorous and accurate. This may alleviate any concerns that the data

have been selected rather than simply reported with all data used for averages, P values etc. The same criticism may apply to supplementary Table 2 and Table S5 for FASN. Fig 4B indicates that FASN is identified in all 3 data sets but not in all experiments for one of the data sets. Could the authors also consider a more detailed legend for the supplementary tables especially indicating exactly how the replicates are averaged and how all the NA values are considered? Perhaps a definition of NA would be helpful as well?

Reply: We thank the Reviewer's comments and revised these sentences according to the Reviewer's suggestion. We considered NAs in the table as not available for quantification based on results of MaxQuant (proteome and ubiquitinome) and Mascot (interactome). Thus, NA means the protein was not identified or didn't satisfy the criteria. Missing value is the nature of mass spectrometry-based proteomics data, which is impossible to be avoided in current technical platforms. Therefore, we used the SILAC-based approach, which is generally considered to be reliable and superior for cellular proteome quantification compared to other methods. The ratio appearing in multiple replicates were averaged by arithmetic mean, which is commonly used to present the quantitative proteomics data (Mol Cell Proteomics. 2018 Feb;17(2):321-334; Mol Cell Proteomics. 2016 Jun;15(6):2093-107; Cell. 2011 Oct 14;147(2):459-74.).

In Fig 4B, FASN was identified in all the replicates of the proteome profiling experiments. In each replicate of the ubiquitinome experiments, some FASN ubiquitination sites were identified. Together, we identified 13 ubiquitination sites. Among them, 4 of the 13 sites were identified in all replicates. In the interactome experiments, FASN was identified in

experiments 1 and 2 of Co-IP experiments. We wish to pointed out that these results from mass spectrometry analysis were largely validated by our independent Western blot analysis in Supplementary Figure 4 for protein expression, Fig. 5E for ubiquitination, and Fig. 5A for Co-IP.

According to the reviewer's suggestion, we added more information to the legend of the tables. In addition, we added the SEM into the tables to more clearly present our data.

The authors now may also wish to consider if all the other experiments in the paper should have n=3 biological replicates with possibly quantification of western blots and assurance that the signal is linear etc. One example is Figs7C that may need more experiments and quantification.

Reply: We thank the Reviewers' comments. More biological replicates could yield statistically more robust conclusion for each individual experiment. Nevertheless, cross-validation between independent experiments could often be more critical to support the conclusion of biological studies, because each experiment can provide complementary/orthogonal evidences to support the result. We carried out 2 biological replicates for some of the molecular biology and animal experiments, such as some Western blot analyses. Nevertheless, as we explained above, all our main results were indeed validated by several independent experiments, such as overexpression vs knockdown, WT vs catalytically inactive mutant, DMSO vs small molecule inhibitor. We believe the reliable results and conclusions can be obtained from these independent experiments by considering them together.

We also wish to point out that the purpose for biological replication

experiment in different model system could be different. In various cases in our study, we paid more attention to the qualitative result rather than quantitative result. For example, in our USP14 KD experiment, it is difficult to achieve a linear result from Western blot because of the intrinsic difference of the knockdown efficiency of USP14 in each shRNA transfected replicate. Nevertheless, our data clearly showed that USP14 had been efficiently knocked down in both of the two replicates (Fig. 7a). Indeed, as we also explained in our previous response, representing 2 or 3 lanes of western blots is a widely accepted common practice to present the data for animal model studies of metabolic disorders in the literatures (Wang PX, et al. *Nat Med.* 2017; 23:439-449; Sun S, et al. *Nat Cell Biol.* 2015; 17:1546-1555; Ma X, et al. *Cell Metab.* 2015; 22:695-708). In addition, as suggested by the Reviewer, we performed additional experiments using 3 samples for Fig. S7C.

The authors are well aware of the current trend away from histograms and SEM to box plots and this may also be considered.

Response: We thank the reviewer's comments. As the reviewer suggested, there is a trend to display the results in box plots for large scale samples. However, in most of our experiments, the data points are between 2 to 4. We think box plot is a better way to present the data with larger number of data points. For example, the default minimum number of data points for the popular web-tool for box plots, BoxPlotR (*Nature Methods* 11, 121–122, 2014), is five. Based on the reviewer's suggestion, in the revised manuscript, we added SEM in Supplementary Table 1 besides the averages and P values, so as to provide more information of the data.

Reviewer #4 (Remarks to the Author):

Under review is the revised manuscript by Liu et al., entitled "Proteome-wide analysis of ubiquitin-specific protease 14 substrates: role as a key modulator of liver steatosis via stabilizing fatty acid synthase in obese mice".

Despite the authors have responded to all the comments, there are still few concerns that need to be addressed:

1- The authors claim that the fragments mass tolerance used in the MS database search was set to 0.5 Da. After inspection of the mqpar.xml file submitted by the authors to the repository iProX, it looks like they have reported the wrong value. As the MS2 spectra have been acquired in high-resolution, the fragments mass tolerance used by MaxQuant was 20 ppm. Please, change the text accordingly.

Reply: We appreciate the reviewer's careful inspection of our uploaded proteomics data. We are sorry for this ambiguous description. As we explained in the previous response letter, we used the default MaxQuant parameters to analyze the SILAC-based mass spectrometry data of proteome and ubiquitinome. For proteome profiling analysis, we carried out high-resolution HCD fragmentation for MS/MS to increase the peptide identification confidence. For ubiquitinome analysis, we carried out low-resolution HCD fragmentation for MS/MS to increase the detection sensitivity of low abundant ubiquitinated peptides in the cellular proteome. We are sorry that we only gave a brief and abbreviated description in the method section, which makes the

inconsistency. We added more detailed information in the method section and the iProX repository, as follows: We used two mass spectrometers to acquire the data (Orbitrap Q Exactive mass spectrometer and Orbitrap Fusion mass spectrometer). The proteome data were acquired by Orbitrap Q Exactive with high resolution MS2 spectrum and the ubiquitinome and interactome data were acquired by Orbitrap Fusion with low resolution MS2 spectrum. The precursor mass tolerance was 10 ppm for Mascot software and 20 ppm for Maxquant software; and the fragments mass tolerance was 20 ppm for the proteome Data and 0.5 Da for the ubiquitinome and interactome data.

2- On the same note, as no Mascot .dat files have been submitted to the repository, what was the value used in the Mascot search? 0.5 Da? If so, such a relaxing fragments m/z error window raises the chance for random match and weakens protein identification accuracy. Please, clarify and submit to iProx also the mascot output .dat files. A more appropriate mass tolerance for Orbitrap MS/MS spectra is in the range of 10-20 ppm or 0.05-0.02 Da.

Reply: We appreciate the reviewer's comments. For the same reason as the ubiquitinome data, we carried out low resolution MS/MS analysis of Co-IP interactome data to increase the detection sensitivity. The interactome MS data was analyzed by Mascot, because this dataset was not labeled by heavy isotope amino acids and also because it is in small size. We added more detailed information in the method section. We also uploaded the dat files to the iProx.

3- The iProX submission looks incomplete. The mqpar.xml for the

ubiquitome analysis is missing. Please, submit also the unprocessed .txt files output of MaxQuant. Again, without such information/file it is not possible to fully evaluate the quality of the MS (data) analysis

Reply: We appreciate the reviewer's comments. We have added the mqpar.xml of the ubiquitome analysis to iProx. In addition, we added the unprocessed.txt files of the proteome data and interactome data to iProx.

4- 8.0 m urea should be 8.0 M urea, 15 mm iodoacetamide should be 15 mM iodoacetamide... please use the right symbol to indicate molar concentration (M) throughout the manuscript.

Reply: We appreciate the comment. We have revised these sentences according to the reviewer's suggestion.

Reviewer #2 (Remarks to the Author):

Comments are in a Word file attached.

Reviewer #3 (Remarks to the Author):

The authors may not have adequately addressed the problem in data analysis and statistics.

This is exemplified in supplementary table 1

VLDLR 0.82 0.84 NA NA 0.83 0.01 0.03

There are two experiments that show NA for VLDLR yet the average is based only on the first two numbers of 0.82 and 0.84 with an average of 0.83, a standard deviation of the mean of 0.01. Ignoring the 2 experiments with NA may not be an accurate representation of the data regardless of the explanation and seems contrary to the definition of what is the standard deviation of the mean.

Could the authors correct these errors in SEM, p values etc throughout the supplementary tables and Figures where all data may be considered for inclusion including the experiments that gave no detectable observation?

The authors agree in their rebuttal that having biological replicates could give “statistically more robust conclusion”. Hence, it may be prudent to await more biological replicates. If each orthogonal experiment is not sound statistically then their consideration together may not be valid.

It is unclear if the conclusions of the paper are supported by the data.

Reviewer #4 (Remarks to the Author):

The authors have responded well to all my comments. The manuscript and the associated data set have now been substantially improved. It is now suitable for publication.

Reviewer #2 (Remarks to the Author):

Thanks for these additional experiments. However, ALT and AST levels are surprisingly equal (usually AST are higher than ALT concentrations) and the legends of the ALT and AST units should be added on both Supp Fig 5A and Supp Fig 8A. In addition, it is also puzzling that the saline control is at 40 in supp Fig5 and at 260 in supp Fig8...

Response: We appreciate the comment. We performed additional experiments using a new assay kit (ALT, TE0121-100T, AST, TE0131-100T, Beijing Leagene Biotechnology) as suggested by the reviewer. As shown in the new Supplementary Fig. 5A and 8A, our new experiment demonstrated that AST were indeed higher than ALT. We believe the discrepancy of our previous data could be the expiration of our previous kits which had been used for a long time. The legends of the ALT and AST units have been added to the Supplementary Fig. 5A and 8A accordingly. The difference of saline control between Supplementary Fig 5 and Fig 8 is due to the different mouse models that were used. The saline control mice in Fig 5A were normal C57 mice, but db/db mice in Figure 8A. Due to the severe obesity, diabetes, and fatty liver in db/db mice, the ALT and AST were higher than the normal ones. This phenomenon has been well-documented by literatures (Wang Q, et al. *Hepatology*. 2009; 49:1166-1175; Liu J, et al. *Cell*. 2016; 167:1052-1066.)

Information added in the « Methods » section is insufficient. The authors must supply a table with clinical (fasting glycemia, HOMA-IR, etc ...) and biochemical (ALT, AST, % of hepatocytes with steatosis, etc...)

characteristics of the patients included in the study, to clearly show the basis of the criteria of inclusion in the control and NAFLD groups. The patients were live donors for liver transplantation. Importantly, steatosis in the liver graft is a risk of graft failure, thus questioning the choice of this human cohort for the study. In Fig 1G the authors may indicate with a color code the patients included in the « normal » and the NAFLD group.

Response: We appreciate the comment. The clinical characteristics of study subjects are shown in Supplementary Table 7. We agree with the reviewer that fatty liver is indeed a risk factor for graft failure. However, many liver transplant donors in China are fatty liver due to the insufficiency of normal donors. Our study followed the ethical compliance approved by the Human Research Ethics Committee of Zhongshan Hospital, Fudan University. According to the Reviewer's suggestion, we have modified Figure 1G in which the black dots represent normal and black triangles represent NAFLD.

Thank you for completing this part of the « Methods » section. However, you may change the subtitle of the paragraph which is no more only « Adenovirus preparation » since now it includes STZ and 3-MA protocols.

Response: We appreciate the comment. We have merged them into one as follow:

Animal experiments and adenovirus preparation

Male C57BL/6, and *db/db* mice aged 8 weeks were purchased from the Shanghai Laboratory Animal Company (SLAC) and Nanjing Biomedical Research Institute of Nanjing University, respectively. All mice were housed at 21°C ± 1°C with humidity of 55% ± 10% and a 12-hour light/12-hour dark cycle. HFD-induced obese mice were maintained with free access to high-fat chow (D12492; Research Diets) for 12 weeks. Adenovirus-expressing murine USP14 was constructed by Invitrogen (Shanghai, China) with a full-length USP14 complementary DNA (cDNA) coding sequence. Overexpression of hepatic USP14 was achieved by means of tail-vein injection of Ad-USP14 (2×10^9 plaque-forming units) in normal C57BL/6 mice. To determine whether the role of USP14 is dependent on insulin, C57BL/6 were treated with streptozotocin (STZ, 240 mg/kg). 14 days later, mice (blood glucose levels exceeding 300 mg/dL) were administrated with adenovirus containing USP14 or GFP via tail vein injection. 10 days later, mice were sacrificed for analysis. To inhibit autophagy, some mice overexpressing USP14 were administrated with 3-MA (30 mg/kg/day, i.p.) for 5 days. To silence USP14 expression in *db/db* mice, two adenoviruses expressing USP14 shRNA (1×10^9 plaque-forming units) were generated using pAD_BLOCK_IT_DEST vectors (Invitrogen, Grand Island, New York, USA). All viruses were purified by the cesium chloride method and dialyzed in PBS containing 10% glycerol prior to animal injection. The animal protocol was reviewed and approved by the Animal Care Committee of Zhongshan Hospital.

Human samples

For analysis of hepatic UPS14 expression and TG contents, liver biopsy was performed in those subjects who donated their partial livers for liver transplantation. The subjects were screened with physical examination

and type B ultrasonography. The clinical characteristics of study subjects are shown in Supplementary Table 7. The human study was approved by the Human Research Ethics Committee of Zhongshan Hospital.

Thank you for the modification. « endogenous » can be removed in « endogenous antibodies ».

Response: We appreciate the comment. We have removed these words throughout the whole manuscript.

Reviewer #3 (Remarks to the Author):

The authors may not have adequately addressed the problem in data analysis and statistics.

This is exemplified in supplementary table 1

VLDLR 0.82 0.84 NA NA 0.83 0.01 0.03

There are two experiments that show NA for VLDLR yet the average is based only on the first two numbers of 0.82 and 0.84 with an average of 0.83, a standard deviation of the mean of 0.01. Ignoring the 2 experiments with NA may not be an accurate representation of the data regardless of the explanation and seems contrary to the definition of what is the standard deviation of the mean.

Could the authors correct these errors in SEM, p values etc throughout the supplementary tables and Figures where all data may be considered

for inclusion including the experiments that gave no detectable observation?”

Response: We thank the reviewer’s careful consideration. As we explained in our previous response letter, this processing procedure is probably the most common way to present the quantitative proteomics data (Mol Cell Proteomics. 2018 Feb;17(2):321-334; Mol Cell Proteomics. 2016 Jun;15(6):2093-107; Cell. 2011 Oct 14;147(2):459-74.). Technically, it is impossible to include the missing value for statistical analysis. It is not correct to consider the missing values as either “0” or “infinite”. Indeed, in statistics, missing values are missing randomly so there is no way to calculate the mean and SEM with these values without ignoring the NAs. More importantly, removing the NAs will reduce the sample sizes but, mathematically, it still gives a fair estimation of the mean and SEM. So the resulting p value is still trustable although the statistical power has been reduced. This is the common way implemented by many statistical softwares, such as R, SPSS, and SRS.

The authors agree in their rebuttal that having biological replicates could give “statistically more robust conclusion”. Hence, it may be prudent to await more biological replicates. If each orthogonal experiment is not sound statistically then their consideration together may not be valid.

It is unclear if the conclusions of the paper are supported by the data.

Response: We thank the Reviewer's comments. Indeed, almost all of our experiments were conducted in multiple biological replicates. As explained in the figure legends, we used 6-8 mouse tissue samples for Western blot and other biological experiments. We are sorry that, in our previous response, we misunderstood the Reviewer's meaning as "to present all the replicate samples in ONE SDS-PAGE gel". Our multiple replicates were run in separate SDS-PAGE gels because there are too many samples to be included in one gel and also because representing 2 lanes of western blots is a widely-accepted way for animal model studies of metabolic disorders in many journals including *Nature Communications* (Porteiro et al. Nat Commun. 2017; 8: 15111; Liu et al., Nat Commun. 2017; 8: 14824; Kim et al. Nat Commun. 2017; 8: 2247; Wang PX, et al. Nat Med. 2017; 23:439-449; Sun S, et al. Nat Cell Biol. 2015; 17:1546-1555; Ma X, et al. Cell Metab. 2015; 22:695-708). The Western blot of additional biological replicates were shown below. In addition, we presented our data in boxplot with quantification of western blots in our revised manuscript, including Fig.1B, Fig.1D, Fig. 1F, Fig. 5C, Fig. 6A, Fig. 7A, Supplementary Fig. 6B, Supplementary Fig. 7C, and Supplementary Fig. 8C. All the results showed the consistent statistical difference, which supported all our conclusions.

Figure 1B

Figure 1D

Figure 6A

Figure 7A

Western blots of additional biological replicates

Reviewer #2 (Remarks to the Author):

Thank you for these additional modifications.

There is no more comments on the manuscript.

Reviewer #3 (Remarks to the Author):

The authors may not have considered some of the recommendations for the manuscript and data. In supplementary Table 1, several observations indicate NA yet these data are all ignored in the statistical assessment. This may vitiate supplementary table 1 and Figure 2. The authors indicate in the results that ACLY is down regulated in USP14 KD cells (right side of Fig3C) yet ACLY is not found in supplementary Table 1.

In supplementary Table 5 FASN shows in one of 3 experiments NA yet this is not considered and therefore this strategy may be vitiated.

The use of IgG as a control for Fig5 may be insufficient since there is no IP under such conditions whereas the IP of USP14 may have non specific associated proteins. There are other controls that may be considered. In Fig5C there are 2 lanes for each condition that do not seem to be explained. For Fig5E since there is expected to be different levels of FASN how has this been normalized? For Figs6A, 7A what are the 2 lanes for each condition?

Reviewer #3's comments:

1. The authors may not have considered some of the recommendations for the manuscript and data. In supplementary Table 1, several observations indicate NA yet these data are all ignored in the statistical assessment. This may vitiate supplementary table 1 and Figure 2.

Response: Based on the statistical point of view, we are afraid we could not agree such method will vitiate our conclusion.

First, in technique, missing values in our dataset are missing randomly so there is no obvious bias will be generated by ignoring the NAs. Missing-data imputation method to obtain a higher statistical power for small sample size (such as 4 replicates in our data shown in Supplementary Table 1) is unreliable and not commonly accepted. For each protein, when one or two NA observations are not considered into statistics, the sample size n will be smaller, but it still gives a statistically fair estimation of the mean and SEM. Such calculation will reduce its statistical power (i.e. the calculated p value will be bigger), but the resulting p value is still statistically significant and trustable. Therefore, in our case, the risk of false negative (Type II error) need be considered (some proteins with true differential-expression will not be reported), but the reliability of positive result will not be vitiated when a strict threshold of Type I error rate (p -value) is chosen in prior (i.e., $p < 0.05$ in our data).

Second, as we have already explained in our previous response letters, missing value is the nature of mass spectrometry-based proteomics data, which is impossible to be avoided in current technological platforms. This processing procedure is probably the most common way to present the quantitative proteomics data (*Mol Cell Proteomics*. 2018 Feb;17(2):321-334; *Mol Cell Proteomics*. 2016 Jun;15(6):2093-107; *Cell*. 2011 Oct 14;147(2):459-74).

2. The authors indicate in the results that ACLY is down regulated in USP14 KD cells (right side of Fig3C) yet ACLY is not found in supplementary Table 1.

Response: The reviewer likely misunderstood the results between Figure 3C

and supplementary Table 1. As we explained in our manuscript and also in our previous response letter, we defined proteins with P-value < 0.05 and ≥ 1.2 fold-of-change as the significantly up- or down-regulated proteins and reported in supplementary Table 1, which were used for further bioinformatics analysis. In Figure 3C, we presented all the proteins and ubiquitinated sites identified in our proteomics analysis to show the global picture of the fatty acid and TCA cycle pathways in response to USP14 KD. As clearly shown in Figure 3C, although it showed statistical differential expression between wide type and USP14 KD cells, the ratio change of ACLY (i.e., 0.86) is smaller than 1.2 fold, which did not fit the criteria in supplementary Table 1. Therefore, we did not include this protein in supplementary Table 1.

3. In supplementary Table 5 FASN shows in one of 3 experiments NA yet this is not considered and therefore this strategy may be vitiated.

Response: The reviewer likely misinterpreted the data in supplementary Table 5. As shown in supplementary Table 5 and below, FASN were identified from two of three biological replicates.

prot_acc	prot_desc	Gene Name	emPAI					
			1		2		3	
			control	overexpress	control	overexpress	control	overexpress
P49327	Fatty acid synthase OS=Hor	FASN	NA	0.10	0.01	0.18	NA	NA

As we have already given a detailed response to co-immunoprecipitation (Co-IP) experiment in our first-round of response letter, Co-IP is probably the most widely used classical method for the identification of interacting proteins and a critical step toward the characterization of the substrates of an enzyme (*Nature*. 2012 Jan 5;481(7379):90-3; *Nature*. 2017 Jun 22;546(7659):554-558), although relatively low reproducibility is a well-recognized limitation of this approach. Based on the biochemical point of view, we feel that this reviewer's opinion that considering this data alone to draw a conclusion, but apart from other independent complementary experiments for cross-validation, is not very reasonable.

As we previously explained, given the relatively low expression of some USP14 substrates, the protein loss during purification process and transient/weak interaction between proteins, it was quite common that not every USP14-interacting protein was successfully recovered by all the three

replicates. A previous study led by Drs. Wade Harper and Steven Gygi, two pioneers of protein interactome and ubiquitinome studies, reported that biological replicates only showed a 65% overlap with each other and a 35% overlap with EGFP control complexes (Fig 2C, Cell. 2009; 138:389-403). They also argued that reproducibility itself does not accurately specify a candidate interactor (Fig 2E, Cell. 2009; 138:389-403). Indeed, in two recent interactome studies (Cell. 2015; 162:425-440; Nature. 2017; 545:505-509), they only carried out technical replicates rather than additional biological replicates.

As we also previously clarified, to address the reproducibility issue of co-IP experiment for USP14 substrate identification, we carried out additional independent experiments and used additional criteria as follows:

First, we carried out 3 biological replicates of Co-IP experiment.

Second, we further used a five-fold change of protein quantification emPAI score between the overexpress (bait) and control IP samples as a cutoff and in at least two of the three biological replicates as potential interaction partners of USP14 to further remove low confident/non-specific protein interactor candidates (Mol Cell Proteomics. 2005 Sep;4(9):1265-72; Mol Cell Proteomics. 2009 Dec;8(12):2770-7.; Nucleic Acids Res. 2018 Jan 25;46(2):823-839). We wish to point out that the reproducibility of our experiment was further supported that we still identified 21 of the 25 previously USP14 reported interacting proteins (Cell. 2009; 138:389-403).

Third, we considered the overlap from the two of the three independent experiment datasets (proteome, ubiquitome and interactome) as possible USP14 substrate candidate rather than the single co-IP data alone.

Forth, we carried out independent Western blot analyses to validate USP14 substrate candidates identified from the above mass spectrometry-based dataset. To validate FASN as a USP14 substrate, we carried out Western blot in Supplementary Figure 4 for protein expression, Fig. 5E for ubiquitination, and Fig. 5A for IP.

Together, we think these multiple independent cross-validation experiments well supported FASN as a *bona fide* USP14 substrate.

4. The use of IgG as a control for Fig5 may be insufficient since there is no IP under such conditions whereas the IP of USP14 may have non specific

associated proteins. There are other controls that may be considered.

Response: The reviewer may misunderstand use of IgG as a control. We did carry out IP experiment using IgG as a negative control to eliminate the non-specific binding background, which is the most classical biochemical experiment to detect the protein-protein interaction from endogenous samples (Figure 4A, Shin HJ, et al. *Nature*. 2016;23;534; Figure 4C, Wang Y, et al. *Science*. 2016;7;354). More importantly, the purpose of this experiment was to further independently confirm interactome data from Co-IP/mass spectrometry analysis, which used exogenous Flag-tagged USP14 as a bait. Again, similar to Question #3, our conclusion was further well supported by other cross-validation independent experiment datasets (proteome, ubiquitome and interactome).

5. In Fig5C there are 2 lanes for each condition that do not seem to be explained.

Response: As we clarified in all rounds of our previous response letters, 2 lanes are the representative two biological replicates. Representing 2 lanes of western blots is a widely-accepted way for animal model studies of metabolic disorders in many journals including Nature Communications (Porteiro et al. *Nat Commun*. 2017; 8: 15111; Liu et al., *Nat Commun*. 2017; 8: 14824; Kim et al. *Nat Commun*. 2017; 8: 2247; Wang PX, et al. *Nat Med*. 2017; 23:439-449; Sun S, et al. *Nat Cell Biol*. 2015; 17:1546-1555; Ma X, et al. *Cell Metab*. 2015; 22:695-708). To clarify this more clearly, we added the explanation into the figure legends as “The two lanes were two representative biological replicates. Each lane represents the tissue protein sample from one mouse.”

6. For Fig5E since there is expected to be different levels of FASN how has this been normalized?

Response: We used IP-Western blot assay to independently confirm the ubiquitination status of FASN in the absent of USP14 from our mass spectrometry-based ubiquitinome data. Similar to Question #4, this is a

classical and widely-accepted biochemical experimental design to analyze ubiquitination of protein (Figure 4B, Shin HJ, et al. *Nature*. 2016;23;534; F Fig4C and Fig5A, Koliopoulos MG, et al. *Nat Commun*. 2018 8;9;1820). It is technically impossible to correctly normalize the ubiquitination of FASN as it was degraded and smeared in the SDS PAGE in the USP14 knockdown sample. This experiment was for qualitative validation purpose but not for quantitative purpose, which clearly validated the result from our mass spectrometry data and supported our conclusion that ubiquitination level of FASN was higher after USP14 knockdown. In addition, the normalized ubiquitination status of USP14 has already been described in our ubiquitome dataset.

We believe that our biological conclusion on FASN ubiquitination regulated by UPS14 is reliable, which is supported by multiple independent experiments.

7. For Figs6A, 7A what are the 2 lanes for each condition?

Response: The same to Question #5, 2 lanes are the representative two biological replicates as we explained in all our previous responses. To clarify this more clearly, we added the explanation into the figure legends as “The two lanes were two representative biological replicates. Each lane represents the tissue protein sample from one mouse.”